# Optimal Transport under Group Fairness Constraints

**Linus Bleistein** [* 1]   **Mathieu Dagréou** [* 2]   **Francisco Andrade** [* 2 3]   **Thomas Boudou** [2]   **Aurélien Bellet** [2]

## Abstract

Ensuring fairness in matching algorithms is a key challenge in allocating scarce resources and positions. Focusing on Optimal Transport (OT), we introduce a novel notion of group fairness requiring that the probability of matching two individuals from any two given groups in the OT plan satisfies a predefined target. We first propose a modified Sinkhorn algorithm to compute perfectly fair transport plans efficiently. Since exact fairness can significantly degrade matching quality in practice, we then develop two relaxation strategies. The first one involves solving a penalized OT problem, for which we derive novel finite-sample complexity guarantees. Our second strategy leverages bilevel optimization to learn a ground cost that induces a fair OT solution, and we establish a bound on the deviation of fairness when matching unseen data. Finally, we present empirical results illustrating the performance of our approaches and the trade-off between fairness and transport cost.

## 1. Introduction

Algorithmic matching mechanisms play an increasing role in modern societies, handling the distribution of rare goods by centrally connecting individuals or firms based on their characteristics and preferences rather than through price-driven markets. Examples of such mechanisms include online job recommendations, college admissions systems, dating and ride-hailing apps, and kidney allocation circuits.

An increasing concern is the fairness of such mechanisms: since matchings partially determine individual outcomes (for instance, through marriage or schooling decisions), they

may result in unfair outcomes. For instance, the growing importance of dating apps in contemporary matrimonial dynamics has raised concerns over their role in fostering social and racial marital homogamy (Piketty, 2013; Zheng et al., 2018; Jia et al., 2018; Zhao et al., 2024), a phenomenon often highlighted as a key driver of inequality. Similarly, school admission systems have been criticized for excessively matching students from privileged backgrounds with elite institutions regardless of their academic potential (Hiller & Tercieux, 2012; Fack et al., 2014; Simioni & Steiner, 2022), causing underprivileged students to lose the benefits of elite higher education. More generally, *these concerns arise because matching decisions overly depend on individual characteristics that are highly correlated with attributes defining social groups, such as ethnicity or socioeconomic background.* Such dependence may lead to highly homogeneous pairings, which may conflict with political, philosophical, and legal principles that promote diversity and equality of opportunity. Algorithmic decisions systematically disadvantaging certain groups are often referred to as *group fairness issues*.

Group fairness has been extensively studied in supervised and unsupervised learning (Barocas et al., 2023). In supervised learning, where an algorithm predicts an unknown outcome $Y$ from features $X$, one example of group fairness is demographic parity (Calders et al., 2009), which requires that the prediction $f(X)$ be independent of a protected attribute $S$ (such as gender or age). In unsupervised learning, one approach is equality of representation error, which requires that the reconstruction error of a learned latent representation be independent of the sensitive attribute $S$ (Samadi et al., 2018). Other notions of group fairness have also been proposed in both supervised and unsupervised contexts (Barocas et al., 2023).

Extending these fairness notions to matching problems is, however, nontrivial, as matching cannot be directly framed as either a supervised or unsupervised learning task. Most prior work has focused on individual fairness, ensuring that similar individuals receive similar matches (Devic et al., 2023), or on problem-specific notions, such as guaranteeing a minimal participation level in settings where individuals can opt out (Ashlagi & Roth, 2014). We discuss this literature in more details in the related work section.

---

[*]Equal contribution   [1]School of Computer and Communication Sciences, School of Life Sciences, EPFL, Lausanne, Switzerland [2]PreMeDICaL, Inria, Inserm, Idesp, Université de Montpellier [3]HeKA, Inria, Paris. Correspondence to: Linus Bleistein <linus.bleistein@epfl.ch>, Mathieu Dagréou <mathieu.dagreou@inria.fr>, Francisco Andrade <francisco.de-lima-andrade@inria.fr>.

*Proceedings of the $43^{rd}$ International Conference on Machine Learning*, Seoul, South Korea. PMLR 306, 2026. Copyright 2026 by the author(s).

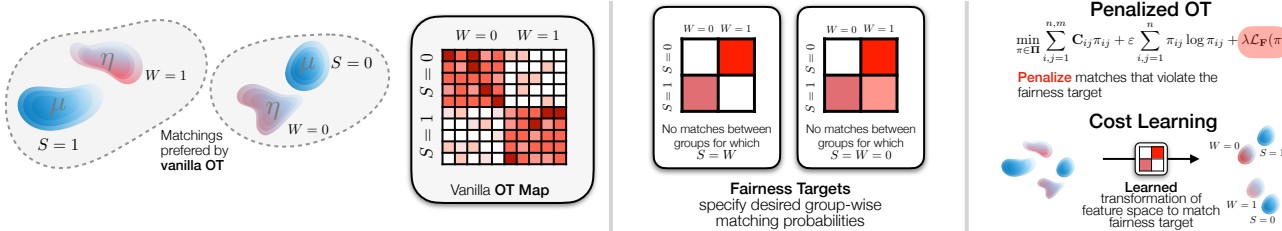

*Figure 1.* Illustration of optimal transport under group fairness constraints. **Left:** Due to the correlation between group attributes ($S$ and $W$) and features, vanilla optimal transport produces transport plans that concentrate most of the mass between matching groups. **Center**: Our novel notion of fairness is based on *fairness targets*, defined as the desired group-level matching probabilities specified by a central planner. **Right**: We propose two approaches for solving this problem: a fairness-penalized optimal transport method, and a cost-learning approach that transforms the input space so that the resulting optimal transport plan aligns as closely as possible with the fairness targets.

**Contributions.** In this work, we study group fairness in matching using the framework of optimal transport (OT). Since the pioneering work of Kantorovitch, OT has been a long standing modeling tool in economics and social sciences (Galichon, 2018) and has gained traction in the last decade as a way of studying complex matching mechanisms in economics (Galichon, 2021; Mastrandrea et al., 2025), logistics and urban network studies (Leite & De Bacco, 2022), demand estimation and pricing (Qu et al., 2025), decision theory, as well as labor and migration economics (Carlier et al., 2023; Hazard & Kitagawa, 2025).

We introduce a novel notion of fairness for optimal transport that allows a central planner—such as a government or regulatory body—to specify the probability that individuals from one group are matched with individuals from another group. This flexible definition can enforce a variety of constraints, including limited homogamy (reducing the likelihood that individuals with the same sensitive attributes are matched), matching quotas (ensuring that a certain percentage of a minority group receives desirable opportunities), or broader goals of social diversity. We summarize our contributions below, with an overview provided in Figure 1.

1. We introduce a novel fair optimal transport problem, and present the `FairSinkhorn` algorithm for computing OT plans under exact fairness constraints.

2. We propose two relaxed approaches. The first adds a penalty to the optimal transport objective to encourage fair plans, yielding a convex optimization problem where the cost–fairness trade-off is easily controlled via the penalty parameter. The second learns a cost function such that the resulting transport plan is fair, allowing straightforward matching of new samples.

3. For the first approach, we provide a novel sample complexity bound for penalized entropic optimal transport, which we believe to be of independent interest. For the cost learning approach, we bound the expected fairness deviation between finite samples and the distribution, holding for any cost in a given parameterized family.

4. We investigate the empirical performance of the proposed approaches through numerical experiments.

Throughout this article, all proofs are deferred to the appendix. Our code is available at https://github.com/LinusBleistein/fair_ot/.

## 2. Related Work

**Fairness in matching algorithms.** Fairness has been studied across a variety of matching problems. A first line of work investigates individual-level fairness in Gale–Shapley–type matching mechanisms (Gale & Shapley, 1962), requiring that similar individuals receive similar outcomes (Karni et al., 2022; Devic et al., 2023). A second line of work focuses on group-level fairness, primarily in the specific case of kidney paired donation (KPD) programs (Ashlagi & Roth, 2014; Dickerson et al., 2014; Zhang et al., 2026; Lobo et al., 2025). Our work departs from this literature in two key ways. First, we do not consider iterative agent-level mechanisms (in which agents are sequentially paired until the match is stable) but instead model matching through (entropic) optimal transport. Second, rather than individual rankings, we assume that both populations are structured by sensitive attributes and impose explicit inter-group mass constraints specifying how much probability mass should flow between subgroups. Another related line of work studies fairness-constrained bipartite matching on graphs $(\mathcal{U} \cup \mathcal{V}, \mathcal{E})$, where edges encode feasible matches (Godsil, 1981; Zdeborová & Mézard, 2006; Noiry et al., 2021). Recent work in this setting includes (Castera et al., 2025), who partition one side of the graph ($\mathcal{U}$ or $\mathcal{V}$) into groups and limit the number of vertices that can be matched from each group, and (Panda et al., 2024), who further impose individual fairness constraints. Our work is fundamentally different from theirs since ($i$) individuals are represented by features rather than graph positions; ($ii$) we allow mass splitting rather than integral matchings; and ($iii$) our formulation is symmetric in the two populations.

**Fairness and optimal transport.** A growing literature uses

optimal transport to obtain or characterize fair learning algorithms (Gordaliza et al., 2019; Gouic et al., 2020; Chzhen et al., 2020; Chiappa et al., 2020; Buyl & Bie, 2022; Hu et al., 2023; Xian et al., 2023; Xiong et al., 2024; Divol & Gaucher, 2024). A central insight is that fair prediction can often be formulated as a Wasserstein barycenter or projection problem (Gouic et al., 2020; Chzhen et al., 2020; Divol & Gaucher, 2024). In this line of work, OT is a tool to enforce fairness in downstream prediction tasks. Our perspective is orthogonal: *we study the fairness of the transport plan itself.* Closest to our problem is Nguyen et al. (2025), who impose fairness on a Wasserstein barycenter problem by enforcing approximate equality of its sliced-Wasserstein distances to the input measures. In contrast, we study the problem of computing transport plans that satisfy explicit mass constraints between groups defined by sensitive attributes, a setting that, to the best of our knowledge, has not been previously considered.

**Constrained and statistical optimal transport.** Our approach builds on constrained optimal transport, which enforces structural properties of transport plans (Courty et al., 2016; Alvarez-Melis et al., 2018; Blondel et al., 2018; Paty & Cuturi, 2019; Scetbon et al., 2021; Liu et al., 2023; Manupriya et al., 2024), including bounds on transported mass (Korman & McCann, 2015). We build on and extend the theory of statistical and penalized optimal transport (Rakotomamonjy et al., 2015; Genevay et al., 2019; Rigollet & Stromme, 2025) to derive finite-sample guarantees and to control deviations from the prescribed fairness constraints. We provide an extended related work review in Appendix A.

## 3. Formalizing Group Fairness in OT

### 3.1. Optimal Transport

We consider the entropic optimal transport problem (Peyré et al., 2019; Genevay et al., 2019; Feydy et al., 2019; Pooladian & Niles-Weed, 2021; Keriven, 2023) between two probability distributions $\mu$ and $\eta$ with ground cost $c$:

$$\min_{\pi \in \Pi(\mu,\eta)} \int_{\mathcal{X} \times \mathcal{Y}} c(x,y) \, d\pi(x,y) + \varepsilon \mathbf{KL}\big(\pi | \mu \otimes \eta\big),$$

where the minimum is taken over couplings $\Pi(\mu,\eta)$ of $\mu$ and $\eta$ and where $\mathbf{KL}$ denotes the Kullback-Leibler (KL) divergence (Csiszár, 1975). We focus on the entropic formulation as regularization ensures uniqueness and smoothness of the optimal coupling, which is crucial for efficient computation and gradient-based optimization (Peyré et al., 2019).

In the finite-sample setting, when $\mu$ and $\eta$ are sums of Dirac measures i.e. if $\mu = 1/n \sum_{i=1}^{n} \delta_{\mathbf{x}_i}$ and

$\eta = 1/m \sum_{j=1}^{m} \delta_{\mathbf{y}_j}$, the problem above reduces to

$$\min_{\mathbf{\Pi} \in \Pi} \sum_{i=1}^{n} \sum_{j=1}^{m} \mathbf{\Pi}_{ij} \mathbf{C}_{ij} + \varepsilon \mathbf{KL}\big(\mathbf{\Pi}\big) \ , \qquad (1)$$

where $\Pi := \big\{ \mathbf{\Pi} \in \mathbb{R}_+^{n \times m} \, | \, \mathbf{\Pi} \mathbf{1}_m = 1/n \mathbf{1}_n, \, \mathbf{\Pi}^\top \mathbf{1}_n = 1/m \mathbf{1}_m \big\}$, $\mathbf{KL}\big(\mathbf{\Pi}\big) := \sum_{ij} \mathbf{\Pi}_{ij} \log \mathbf{\Pi}_{ij}$ and $\mathbf{C} \in \mathbb{R}^{n \times m}$ with $\mathbf{C}_{ij} := c(x_i, y_j)$. The KL divergence is strictly convex, and $\Pi$ is a nonempty and bounded set. Therefore, (1) admits a unique minimizer, which we denote by $\mathbf{\Pi}_\varepsilon(\mathbf{C})$.

### 3.2. Fair Optimal Transport

We consider two distributions $\mu \in \mathcal{P}(\mathcal{X} \times \mathcal{S})$ and $\eta \in \mathcal{P}(\mathcal{Y} \times \mathcal{W})$ to be matched, where $\mathcal{X}$ and $\mathcal{Y}$ represent feature spaces while $\mathcal{S}$ and $\mathcal{W}$ correspond to sensitive attributes (e.g., gender, ethnicity or age) defining groups of entities in $\mu$ and $\eta$ respectively. The sets $\mathcal{W}$ and $\mathcal{S}$ are assumed to be finite, and are identified, respectively, with $\{1, \ldots, K_w\}$ and $\{1, \ldots, K_s\}$. To obtain theoretical results on sample complexity, we will need an assumption on the support of the measures, which is standard in statistical optimal transport (Rigollet & Stromme, 2025).

**Assumption 3.1.** $\mu$ and $\eta$ are compactly supported.

We denote by $p_s = \mathbb{P}(S = s)$ and $q_w = \mathbb{P}(W = w)$ the probability of each group $i$ in $\mu$ and $j$ in $\eta$, respectively, and by $\mathbf{p}$ and $\mathbf{q}$ the resulting distributions over $\mathcal{S}$ and $\mathcal{W}$.

**Fairness targets.** Optimal transport imposes structure on the coupling of two probability distributions through the cost function, making it more likely to match points whose transportation cost to one another is low. While this inductive bias is often desirable, it can lead to undesirable outcomes when features in $\mathcal{X}$ and $\mathcal{Y}$ are correlated with the sensitive attributes in $\mathcal{S}$ and $\mathcal{W}$, as the resulting coupling may reflect, preserve, or amplify disparities between the corresponding groups, as illustrated in the following example.

**Example 3.1.** *Consider a school assignment system, in which students have a location $X \in \mathbb{R}^2$ and social status $S \in \{\text{high}, \text{low}\}$, and schools have a location $Y \in \mathbb{R}^2$ and prestige $W \in \{\text{elite}, \text{non-elite}\}$. If high-status students tend to live near elite schools and low-status students near non-elite schools, then optimal transport with Euclidean cost—minimizing the distance between students and schools—will produce highly segregated schools.*

In this example, the block-sparse structure of the optimal transport plan can be seen as a source of unfairness, as the matching will be highly correlated with the social status of the students and the elitist nature of the schools.

To address such issues and define fair transport plans, we are given a *fairness target* $\mathbf{F}$, which is a $K_s \times K_w$ matrix

that specifies, for each pair of groups $(s, w) \in \mathcal{S} \times \mathcal{W}$, the desired probability of matching members of group $s$ with group $w$. To be valid, the matrix $\mathbf{F}$ should itself be a coupling between $\mathbf{p}$ and $\mathbf{q}$: it should be non-negative and satisfy the following constraints for all $(s, w)$:

$$\sum w = 1^{K_w} \mathbf{F}_{sw} = p_s \text{ and } \sum s = 1^{K_s} \mathbf{F}_{sw} = q_w \ .$$

We hence write $\Pi(\mathbf{p}, \mathbf{q})$ the set of fairness targets.

**Example 3.2.** *Consider the segregated schooling system of Example 3.1. Let $p := \mathbb{P}(S = \text{low})$ and $q := \mathbb{P}(W = \text{non-elite})$. A newly appointed city administrator wishes to limit social homogamy within public schools, and hence requires $60\%$ of student with low social status to be matched to elite schools. The fairness target can be written as*

$$\mathbf{F} = \begin{bmatrix} 0.4 \times p & 0.6 \times p \\ q - 0.4 \times p & 1 - q - 0.6 \times p \end{bmatrix} .$$

**Remark 3.1.** Many real-world rules or legislative goals can be naturally mapped to the fairness target concept of our framework. In the US, hiring processes are evaluated under the four-fifths rule, which requires that the selection rate of any demographic group should be at least $80\%$ of that of the group with the highest selection rate.[1] In the French higher education system, several institutions are subject to minimum quotas for scholarship holders.[2] Lastly, at the European level, a recent EU Directive sets a target whereby listed companies should ensure that at least $40\%$ of executive non-executive director positions, or $33\%$ of all director positions, are held by women by 2026.[3]

**Fair optimal transport.** We say that a coupling is $\mathbf{F}$-fair if the amount of mass transported from group $s$ to group $w$ equals the prescribed fairness target $\mathbf{F}_{sw}$. An $\mathbf{F}$-fair optimal transport plan is then an $\mathbf{F}$-fair coupling that minimizes the transport cost among all such couplings.

**Definition 3.3** (**F**-Fair Optimal Transport). *Let $\mu \in \mathcal{P}(\mathcal{X} \times \mathcal{S})$ and $\eta \in \mathcal{P}(\mathcal{Y} \times \mathcal{W})$. An $\mathbf{F}$-fair optimal transport plans between $\mu$ and $\eta$ is defined as*

$$\pi_{c,\varepsilon,\mathbf{F}}^\star \in \underset{\pi \in \Pi_{\mathbf{F}}(\mu, \eta)}{\arg\min} \int c(x, y) \, d\pi(x, y) + \varepsilon \mathbf{KL}(\pi | \mu \otimes \eta) \quad (2)$$

*where the set of $\mathbf{F}$-fair couplings $\Pi_{\mathbf{F}}(\mu, \eta)$ is given by*

$$\Pi_{\mathbf{F}}(\mu, \eta) := \left\{ \pi \in \Pi(\mu, \eta) \mid \forall (s, w), \, \pi_{SW}(s, w) = \mathbf{F}_{sw} \right\}$$

*with $\pi_{SW}$ denoting the induced coupling on $\mathcal{S} \times \mathcal{W}$ obtained from $\pi$ by marginalizing over $x$ and $y$, that is*

$$\pi_{SW}(s, w) := \pi(\mathcal{X} \times \{s\} \times \mathcal{Y} \times \{w\}) \ .$$

[1] Disparate impact (Wikipedia Page) (Accessed: 2026-05-08)
[2] Parcoursup 2026 : Encourager la mobilité sociale et géographique sur Parcoursup (Accessed: 2026-05-08)
[3] Directive (EU) 2022/2381 (Accessed: 2026-05-08)

In the finite-sample case, where we have access to two datasets $(\mathbf{x}_i, \mathbf{s}_i)_{i=1}^n \in \mathcal{X} \times \mathcal{S}$ and $(\mathbf{y}_i, \mathbf{w}_i)_{i=1}^m \in \mathcal{Y} \times \mathcal{W}$ drawn i.i.d. from $\mu$ and $\eta$ respectively, the fair optimal transport problem (2) writes

$$\min_{\mathbf{\Pi} \in \Pi} \text{Tr}\left[\mathbf{\Pi}^\top \mathbf{C}\right] + \varepsilon \mathbf{KL}(\mathbf{\Pi})$$

$$\text{s.t. } \forall (s, w), \text{Tr}\left[\mathbf{\Pi}^\top \mathbf{B}_{sw}\right] = \sum_{i|\mathbf{s}_i=s} \sum_{j|\mathbf{w}_j=w} \mathbf{\Pi}_{ij} = \mathbf{F}_{sw} \quad (3)$$

where $\mathbf{B}_{sw} := \left(\mathbb{1}_{\mathbf{s}_i=s} \mathbb{1}_{\mathbf{w}_j=w}\right)_{i,j} \in \{0, 1\}^{n \times m}$.

The following proposition ensures that the fair OT problem is well defined.

> **Proposition 3.2.** Under Assumption 3.1, for any target $\mathbf{F} \in \Pi(\mathbf{p}, \mathbf{q})$, there exists a unique fair OT plan.

**Remark 3.2** (Extension to alternative constraints). Our framework naturally extends to settings in which fairness targets are imposed only on a subset of group pairs, leaving remaining matching probabilities unconstrained, as well as to cost-weighted fairness targets that, for example, balance transport costs across groups.

**Remark 3.3** (From probabilistic to deterministic transport plans). Given a fair entropic transport plan, a deterministic matching can be generated by sampling from the coupling. Our empirical results indicate that, for sufficiently large samples, the resulting matching exhibits nearly the same level of fairness (see Figure 7b in Appendix D.4).

### 3.3. Solving Fair Optimal Transport

We focus on solving the finite-sample version (3) of the fair optimal transport problem. Importantly, the fairness constraints are linear in the transport plan, which allows us to leverage the dual formulation and obtain the following result. In what follows, $\odot$ and $\oslash$ denote the elementwise product and division of matrices or vectors, respectively.

> **Proposition 3.3.** There exists $\mathbf{f} \in \mathbb{R}^n$, $\mathbf{g} \in \mathbb{R}^m$ and $\mathbf{h} = [h_{sw}]_{sw} \in \mathbb{R}^{K_s \times K_w}$ such that the solution to Problem (3) has the form
>
> $$\mathbf{\Pi} = \text{diag}\left(e^{\mathbf{f}/\varepsilon}\right) \left(\mathbf{K} \odot \mathbf{H}\right) \text{diag}\left(e^{\mathbf{g}/\varepsilon}\right) \ ,$$
>
> where $\mathbf{K} := \left[e^{-\mathbf{C}_{ij}/\varepsilon}\right]_{ij}$ and $\mathbf{H} := \sum_{sw} e^{h_{sw}/\varepsilon} \mathbf{B}_{sw}$.

This result mirrors the foundational insight behind the Sinkhorn algorithm and shows that the fairness constraint can be incorporated via a simple additional projection step. This leads to our modified Sinkhorn algorithm, which we call `FairSinkhorn`, presented in Algorithm 1 in which the function $\Phi$ is defined as

$$[\Phi(\mathbf{u}, \mathbf{v})]_{sw} := \sum_{i|s_i=s} \sum_{j|w_j=w} \mathbf{u}_i \mathbf{v}_j \mathbf{K}_{i,j}$$

**Algorithm 1** `FairSinkhorn` algorithm to solve (3)

1: **Inputs:** Cost $\mathbf{C} \in \mathbb{R}^{n \times m}$, $\varepsilon > 0$, fairness target $\mathbf{F} = (\mathbf{F}_{sw})_{sw} \in [0,1]^{K_s \times K_w}$, number of iterations $T$, initializations $\mathbf{u}^{(0)} \in \mathbb{R}^n$, $\mathbf{v}^{(0)} \in \mathbb{R}^m$, and $\mathbf{L}^{(0)} \in \mathbb{R}^{K_s \times K_w}$.
2: $\mathbf{K} \leftarrow e^{-\mathbf{C}/\varepsilon - 1}$
3: $\mathbf{T}^{(0)} \leftarrow \sum_{sw} \ell_{sw}^{(0)} \mathbf{B}_{sw}$ with $\ell_{sw}^{(0)} = \left(\mathbf{L}^{(0)}\right)_{sw}$
4: **for** $t = 0, \ldots, T-1$ **do**
5: $\quad \mathbf{u}^{(t+1)} \leftarrow n^{-1}\mathbf{1} \oslash \left((\mathbf{K}\odot\mathbf{T}^{(t)})\mathbf{v}^{(t)}\right)$
6: $\quad \mathbf{v}^{(t+1)} \leftarrow m^{-1}\mathbf{1} \oslash \left((\mathbf{K}\odot\mathbf{T}^{(t)})^{\top}\mathbf{u}^{(t+1)}\right)$
7: $\quad \color{blue}\mathbf{L}^{(t+1)} \leftarrow \mathbf{F} \oslash \Phi(\mathbf{u}^{(t+1)}, \mathbf{v}^{(t+1)})$
8: $\quad \color{blue}\mathbf{T}^{(t+1)} \leftarrow \sum_{sw} \ell_{sw}^{(t+1)} \mathbf{B}_{sw}$ with $\ell_{sw}^{(t+1)} = \left(\mathbf{L}^{(t+1)}\right)_{sw}$
9: **end for**
10: **Return:** $\mathbf{\Pi} = \mathrm{diag}(\mathbf{u}^{(T)})(\mathbf{K}\odot\mathbf{T}^{(T)})\mathrm{diag}(\mathbf{v}^{(T)})$

The additional steps relative to the original Sinkhorn algorithm are highlighted in blue. We empirically compare the convergence speed of `FairSinkhorn` and regular `Sinkhorn` in Appendix D.5 and show that they have qualitatively similar rates.

## 4. Two Strategies for Approximately Fair OT

`FairSinkhorn` enforces *exact fairness*, which can substantially increase transport cost, sometimes beyond what is acceptable in practice. This *price of fairness* has been studied in operations research for resource allocation problems (Bertsimas et al., 2011) and also arises in supervised learning, where enforcing exact group fairness constraints may sacrifice predictive accuracy (Zhao & Gordon, 2019).

Motivated by these observations, in this section we study approximately fair OT formulations, replacing the linear fairness constraints in (3) with a relaxed condition that allows slight deviations from the fairness target while still controlling the total fairness violation:

$$\mathcal{L}_{\mathbf{F}}(\mathbf{\Pi}) := \sum_{(s,w)\in\mathcal{S}\times\mathcal{W}} \left(\mathrm{Tr}[\mathbf{\Pi}^{\top}\mathbf{B}_{sw}] - \mathbf{F}_{sw}\right)^2 \leq \rho, \quad (4)$$

where $\rho \geq 0$ is a tolerance level. To find OT plans under this relaxed constraint, we propose two strategies.

**Remark 4.1** (Extension to range-based constraints). Our framework can also be extended to accommodate range-based fairness constraints, where the fairness target requires the probability of matches between groups to lie within a specified interval. Instead of penalizing the squared difference in (4), one could for instance use a margin-based loss that only penalizes violations outside the interval.

### 4.1. Fairness-Penalized OT

Our first strategy is a direct penalization approach. We incorporate the $\rho$-relaxed fairness constraint (4) into the optimal

transport problem by introducing a Lagrange multiplier, yielding the penalized objective

$$\min_{\mathbf{\Pi}\in\Pi} \mathrm{Tr}[\mathbf{\Pi}^{\top}\mathbf{C}] + \varepsilon\mathbf{KL}(\mathbf{\Pi}) + \lambda\mathcal{L}_{\mathbf{F}}(\mathbf{\Pi}), \quad (5)$$

where $\lambda > 0$ controls the strength of the fairness penalty. Importantly, since $\mathcal{L}_{\mathbf{F}}$ is convex, the objective remains strongly convex, ensuring a unique minimizer. This penalized entropic optimal transport problem can be solved efficiently using a generalized conditional gradient algorithm (Rakotomamonjy et al., 2015). At each iteration, the algorithm linearizes the fairness regularization around the current iterate and solves a subproblem over the feasible set $\Pi$ to find a search direction. A line search then determines a convex combination of the current iterate and this direction, guaranteeing descent. The subproblem at iteration $t$ corresponds to an entropic OT problem with the modified cost $\mathbf{C} + \nabla\mathcal{L}_{\mathbf{F}}(\mathbf{\Pi}^t)$, which is solved efficiently with the Sinkhorn algorithm. This makes the algorithm computationally practical even for large-scale problems.

**Sample complexity of fairness-penalized OT.** We establish a sample complexity bound for the above fairness-penalized optimal transport cost by building upon results from entropic optimal transport (Genevay et al., 2019; Mena & Niles-Weed, 2019). To formalize this, note that the optimization problem in (5) can be defined for arbitrary measures $\mu \in \mathcal{P}(\mathcal{X} \times \mathcal{S})$ and $\eta \in \mathcal{P}(\mathcal{Y} \times \mathcal{W})$ as (see Appendix C.2 for a formal definition)

$$m^{\star}(\mu, \eta) := \min_{\pi\in\Pi(\mu,\eta)} \int_{\mathbb{R}^d\times\mathbb{R}^d} c(x,y)\, d\pi(x,y)$$
$$+ \varepsilon\mathbf{KL}(\pi\|\mu\otimes\eta) + \lambda\mathcal{L}_{\mathbf{F}}(\pi).$$

Let $\mu_n = \sum_{i=1}^n \delta_{x_i}$ and $\eta_n = \sum_{i=1}^n \delta_{y_i}$ where $(x_i)_{1\leq i\leq n}$ and $(y_i)_{1\leq i\leq n}$ are i.i.d. samples drawn from $\mu$ and $\eta$. In Theorem 4.2, we quantify how well we can estimate $m^{\star}(\mu, \eta)$ with $m^{\star}(\mu_n, \eta_n)$ as a function of the sample size $n$. Similar to (Rigollet & Stromme, 2025) and (Genevay et al., 2019), our proof relies on the fact that the measures are compactly supported (Assumption 3.1). We also require the following assumption on the ground cost function, which is standard in the literature (Genevay et al., 2019).

**Assumption 4.1.** The cost function $c : \mathcal{X} \times \mathcal{Y} \to \mathbb{R}_+$ is infinitely differentiable.

**Theorem 4.2.** Assume for simplicity that $m = n$. Under Assumptions 3.1 and 4.1, we have

$$\mathbb{E}\left|m^{\star}(\mu_n, \eta_n) - m^{\star}(\mu, \eta)\right| \lesssim \frac{1}{\sqrt{n}}.$$

Our result matches the $O(n^{-1/2})$ scaling of sample complexity bounds for standard entropic OT (Genevay et al.,

2019; Mena & Niles-Weed, 2019), showing that adding the fairness penalty does not reduce statistical efficiency. Furthermore, our bound inherits exponential dependency in $\varepsilon$ from Genevay (2019) and in $\lambda$ (we provide more details in Appendix C.2). Interestingly, the result can be extended to *generic convex penalties* at the cost of a supplementary $\log(n)$ factor.

*Proof sketch for Theorem 4.2.* Our proof works by finding two adequate random variables $Y_n$ and $Z_n$ such that

$$Y_n - m^\star(\mu, \eta) \leq m^\star(\mu_n, \eta_n) - m^\star(\mu, \eta) \leq Z_n - m^\star(\mu, \eta).$$

*Lower bound.* We obtain $Y_n$ through linearization of the convex penalty, which is guaranteed to yield a proper lower bound through the connection between the optimality conditions of our penalized problem and its linearized counterpart (Rakotomamonjy et al., 2015). This linearized problem can be cast as an entropic OT problem with modified cost, allowing us to leverage existing sample complexity results (Genevay, 2019; Mena & Niles-Weed, 2019).

*Upper bound.* To obtain $Z_n$ we evaluate the fairness-penalized loss at the minimizer $\hat{\pi}_n^\star$ of the linearized problem, that is, we consider

$$
\begin{aligned}
& m^\star(\mu_n, \eta_n) - m^\star(\mu, \eta) \\
& \leq \langle c, \hat{\pi}_n^\star \rangle + \varepsilon \mathbf{KL}(\hat{\pi}_n^\star | \mu_n \otimes \eta_n) + \lambda \mathcal{L}_F(\hat{\pi}_n^\star) - m^\star(\mu, \eta), \quad (6)
\end{aligned}
$$

where the upper bound follows from the fact that $\hat{\pi}_n^\star$ satisfies the constraints defining $m^\star(\mu_n, \eta_n)$. To establish that the expectation of the absolute value of (6) vanishes at the rate of $1/\sqrt{n}$, we again make use of the connection between the optimality conditions of our penalized problem and its linearized counterpart (Rakotomamonjy et al., 2015) to reduce the problem to the sample complexity of entropic OT. We handle the part $\lambda \mathcal{L}_F(\hat{\pi}_n^\star) - m^\star(\mu, \eta)$ through a generalization of Theorem 6 from (Rigollet & Stromme, 2025). $\square$

## 4.2. Fair OT via Cost Learning

The key idea behind our second strategy is to learn a ground cost function $c_\theta$ so that the entropic OT solution $\mathbf{\Pi}(c_\theta)$ it induces minimizes the fairness violation. In other words, rather than modifying the transport plan directly, we reshape the geometry of the OT problem. This perspective leads naturally to a bilevel optimization problem, with an outer level that adjusts the cost parameters and an inner level that solves an OT problem whose solution depends on those parameters. Formally, we consider

$$
\begin{aligned}
& \min_{\theta \in \Theta} \mathcal{L}_{\mathbf{F}}\big(\mathbf{\Pi}_\varepsilon(c_\theta)\big) + \frac{1}{\lambda} \mathscr{D}(c_\theta, c_{\text{base}}) \\
& \text{s.t. } \mathbf{\Pi}_\varepsilon(c_\theta) = \arg\min_{\mathbf{\Pi} \in \Pi} \text{Tr}\big[\mathbf{\Pi}^\top \mathbf{C}_\theta\big] + \varepsilon \mathbf{KL}(\mathbf{\Pi}),
\end{aligned} \quad (7)
$$

where $(c_\theta)_{\theta \in \Theta}$ is a parameterized family of cost functions, $\mathbf{C}_\theta := \big[c_\theta(\mathbf{x}_i, \mathbf{y}_j)\big]_{ij}$, $c_{\text{base}} : \mathcal{X} \times \mathcal{Y} \to \mathbb{R}^+$ is a baseline cost encoding prior knowledge or domain-specific structure, and $\mathscr{D}$ is a discrepancy measure that encourages the learned cost to remain close to the baseline, thereby controlling deviations from the original OT problem.

**Remark 4.2.** This formulation is related to prior work on cost learning in optimal transport (Li et al., 2019; Andrade et al., 2024), which aims to infer a cost function that explains observed matchings or couplings. Here, however, the objective is different: instead of fitting a cost to reproduce a given transport plan, we learn a cost whose induced OT solution optimizes a downstream criterion, namely fairness.

Problem (7) is well posed since the inner entropic OT problem is strongly convex over $\Pi$, yielding a unique solution $\mathbf{\Pi}_\varepsilon(c_\theta)$ for any $\theta$ and thus a well-defined bilevel objective. This bilevel problem can be optimized using gradient-based methods by differentiating through an iterative solver (iterative differentiation) (Bolte et al., 2023; Franceschi et al., 2017; Maclaurin et al., 2015; Pauwels & Vaiter, 2023) or using implicit differentiation to approximate the gradient of the bilevel objective (Dagréou et al., 2022; Dagréou et al., 2024; Eisenberger et al., 2022; Ghadimi & Wang, 2018).

Our approach can accommodate a wide range of parameterizations for $c_\theta$.

**Example 4.1** (Mahalanobis cost)**.** *Assume $\mathcal{X} = \mathcal{Y} = \mathbb{R}^d$. For a PSD matrix $\mathbf{M}$, the Mahalanobis cost $c_{\mathbf{M}}$ is defined by $c_{\mathbf{M}}(x, y) = (x - y)^\top \mathbf{M}(x - y)$. This corresponds to applying a linear transformation to $x$ and $y$ before computing the squared Euclidean distance. A key advantage of this parametrization is its interpretability: the learned matrix $\mathbf{M}$ directly reveals which directions or features in the data are emphasized or downweighted in the transport cost.*

**Example 4.2** (Neural cost)**.** *A more flexible parametrization uses neural networks $\phi_{\theta_1} : \mathcal{X} \to \mathbb{R}^k$ and $\phi_{\theta_2} : \mathcal{Y} \to \mathbb{R}^k$ to embed the inputs, defining the cost as $c_\theta(x, y) = \|\phi_{\theta_1}(x) - \phi_{\theta_2}(y)\|_2^2$. The networks can be trained from scratch or initialized from a pretrained model and fine-tuned for the specific OT task, enabling highly expressive cost functions that capture complex relationships between $x$ and $y$.*

**Fairness on unseen samples.** A key advantage of our cost learning approach is that it produces a reusable cost function, which can be directly applied to match new samples. This naturally raises the question of whether transport plans obtained by solving an optimal transport problem with the learned cost on previously unseen data will maintain a comparable level of fairness.

To address this question, we derive a pointwise deviation bound on the fairness of the transport plan computed from finite samples relative to the population solution, uniformly over the parameterized cost family. Let $\pi_\varepsilon^\star(c_\theta)$ denote the

OT plan on two measures $\mu$ and $\eta$. Our goal is to control the difference between the fairness of this plan, defined as

$$\mathcal{L}_{\mathbf{F}}\big(\pi_\varepsilon^\star(c_\theta)\big)$$
$$:= \sum_{s=1}^{K_s} \sum_{w=1}^{K_w} \Big( \int_{\mathcal{X}\times\{s\}\times\mathcal{Y}\times\{w\}} d\pi_\varepsilon^\star(c_\theta)(x,u,y,t) - \mathbf{F}_{sv} \Big)^2,$$

and the fairness $\mathcal{L}_{\mathbf{F}}\big(\mathbf{\Pi}(c_\theta)\big)$ of the optimal transport plan $\mathbf{\Pi}(c_\theta)$ computed on finite-sample versions $\mu_n$ and $\eta_m$. We will require the following assumption to hold.

> **Assumption 4.3.** There exists a constant $R_\Theta > 0$ s.t.
>
> $$\sup_{\theta\in\Theta} \|c_\theta\|_\infty := \sup_{\theta\in\Theta} \max_{x\in\mathrm{supp}(\mu), y\in\mathrm{supp}(\eta)} c_\theta(x,y) < R_\Theta.$$

This assumption is mildly restrictive. For instance, it holds if $\Theta$ is bounded and the function $(\theta,x,y) \mapsto c_\theta(x,y)$ is continuous on $\Theta\times\mathcal{X}\times\mathcal{Y}$. For instance, for the Mahalanobis cost function (Example 4.1), the parameter space $\Theta$ is a subset of the space of positive definite matrices and we have

$$R_\Theta = \max_{\mathbf{M}\in\Theta} \|\mathbf{M}\|_{\mathrm{F}} \max_{x\in\mathrm{supp}(\mu), y\in\mathrm{supp}(\eta)} \|x-y\|_2$$

which is finite under Assumption 3.1 as long $\Theta$ is bounded. Similarly, for the neural cost function (Example 4.2), provided that the neural networks are $C_\Theta$-Lipschitz, we have

$$R_\Theta = C_\Theta \max_{x\in\mathrm{supp}(\mu), y\in\mathrm{supp}(\eta)} \|x-y\|_2 \ .$$

> **Theorem 4.4.** Assume for simplicity that $n = m$. Under Assumptions 3.1 and 4.3, we have
>
> $$\sup_{\theta\in\Theta} \mathbb{E}\big[\big|\mathcal{L}_{\mathbf{F}}\big(\mathbf{\Pi}_\epsilon(c_\theta)\big) - \mathcal{L}_{\mathbf{F}}\big(\pi_\epsilon^\star(c_\theta)\big)\big|\big] \lesssim \frac{\exp\big(\frac{5R_\Theta}{\varepsilon}\big)}{\sqrt{n}} \ .$$

*Proof sketch for Theorem 4.4.* We generalize Rigollet & Stromme (2025, Theorem 6) to any bounded cost $c_\theta$ by proving that for any test function $\varphi \in L^\infty(\mu\otimes\eta)$ and for any $t > 0$, we have with probability $1 - 18e^{-t^2}$

$$\Big|\int \varphi d(\mathbf{\Pi}_\epsilon(c_\theta) - \pi_\epsilon^\star(c_\theta))\Big| \lesssim \frac{\exp\big(5R_\Theta\epsilon^{-1}\big)\|\varphi\|_{L^\infty(\mu\otimes\eta)} \cdot t}{\sqrt{n}}$$

which yield the following bound in expectation

$$\mathbb{E}\big[\big|\int \varphi d(\mathbf{\Pi}_\epsilon(c_\theta) - \pi_\epsilon^\star(c_\theta))\big|\big] \lesssim \frac{\exp(5R_\Theta\epsilon^{-1})\|\varphi\|_{L^\infty(\mu\otimes\eta)}}{\sqrt{n}}.$$

Then, using $|a^2 - b^2| \le |a+b||a-b|$, we show that

$$|\mathcal{L}_{\mathbf{F}}(\mathbf{\Pi}_\epsilon(c_\theta)) - \mathcal{L}_{\mathbf{F}}(\pi_\epsilon^\star(c_\theta))| \lesssim \sum_{s=1}^{K_s} \sum_{w=1}^{K_w} \Big| \mathrm{Tr}\big(\mathbf{\Pi}_\epsilon(c_\theta)^\top \mathbf{B}_{uv}\big)$$
$$- \int_{\mathcal{X}\times\{s\}\times\mathcal{Y}\times\{w\}} d\pi_\epsilon^\star(c_\theta)(x,u,y,t)\Big|.$$

Then, for a suitable choice of $\varphi_{sw} \in L^\infty(\mu\otimes\eta)$, we have for any $s, w \in [K_s]\times[K_w]$

$$\Big|\mathrm{Tr}\big(\mathbf{\Pi}_\epsilon(c_\theta)^\top \mathbf{B}_{uv}\big) - \int_{\mathcal{X}\times\{s\}\times\mathcal{Y}\times\{w\}} d\pi_\epsilon^\star(c_\theta)(x,u,y,t)\Big|$$
$$= \Big| \int \varphi_{sw} d(\mathbf{\Pi}_\epsilon(c_\theta) - \pi_\epsilon^\star(c_\theta))\Big|.$$

Putting everything together yields the result. $\square$

This bound provides a uniform control over the expected difference between the fairness loss evaluated on the population and its finite-sample approximation. The exponential dependence on $\varepsilon^{-1}$ is unavoidable in the worst case; see Rigollet & Stromme (2025) and Altschuler et al. (2022) for related discussions. Importantly, Theorem 4.4 implies that, for two sufficiently large samples, the transport plan computed with a given cost achieves a comparable level of fairness. In other words, the same cost function can be reused to match new samples with fairness guarantees.

### 4.3. Comparison of Both Approaches

**Convexity.** The penalized OT problem is convex, whereas the cost learning approach involves a bilevel optimization, which is non-convex and may get stuck in local minima.

**Flexibility.** The cost learning approach restricts the transport map to entropic-regularized solutions with a parametric cost. This problem is thus structurally more constrained than the penalized formulation, which allows solutions outside the set of optimal couplings and is therefore more flexible.

**Reusability and interpretability.** Penalized OT requires solving a new problem for each batch of samples. In contrast, a learned cost can be reused to match new samples via a standard OT problem. Moreover, when the cost belongs to an interpretable family, it can yield insight into how specific features contribute to (un)fairness.

## 5. Experiments

### 5.1. Synthetic Experiments

#### 5.1.1. EXPERIMENTAL SETUP

**Data generation.** Building on the illustrating example of segregated school systems (Examples 3.1–3.2), we consider two synthetic problems. For the first one (Figure 2, left), the positions of students ($X$) and schools ($Y$) are drawn according to a mixture of two Gaussian distributions such that privileged students ($S = 1$) are closer to elite schools ($W = 1$), and underprivileged students closer to non-elite schools. For the second one (Figure 2, right), privileged students and elite schools are drawn from a centered Gaussian distribution, while underprivileged students and non-elite schools are sampled on a centered circle of radius 2. In both cases, there are 10 times more schools than students, and sensitive attributes are balanced.

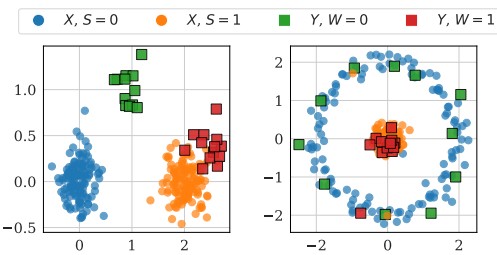

*Figure 2.* Simulated datasets: *Gaussians* (left) and *Circles* (right).

**Fairness target.** We aim to assign approximately 60% of underprivileged students to elite schools, which corresponds to the fairness target

$$\mathbf{F} = \begin{bmatrix} 0.20 & 0.30 \\ 0.28 & 0.22 \end{bmatrix}.$$

**Approaches.** We compare our three approaches: `FairSinkhorn`, penalized OT and cost learning. For cost learning, we evaluate two parameterizations: a Mahalanobis cost (Example 4.1) and a neural cost (Example 4.2), where $\phi_{\theta_1}$ and $\phi_{\theta_2}$ are multilayer perceptrons (MLP). Implementation details are available in Appendix D. We also include vanilla entropic OT with varying $\varepsilon$ as a baseline.

**Metrics.** Following standard practice in fairness research, we analyze the cost–fairness trade-off by visualizing performance in the two-dimensional (*cost*, *fairness*) plane. Cost is measured relative to the transport plan obtained by standard (non-fair) OT with the same entropic regularization, using the difference in total transport cost.

### 5.1.2. COMPARING EXACT AND PENALIZED APPROACHES

To illustrate the need to move from exact group fairness constraints to relaxed ones, we compare, on the Gaussian dataset, the (cost,fairness) trade-off achieved by the original `Sinkhorn` algorithm, our `FairSinkhorn` algorithm (Section 3.2), and our fairness-penalized approach (Section 4.1) for varying values of the fairness penalty. As shown in Figure 3 a., our penalized approach smoothly interpolates between `Sinkhorn`, which attains low transport cost but large fairness violations, and `FairSinkhorn`, which enforces perfect fairness at the expense of higher transport cost. This relaxed formulation thus enables fine-grained control over the fairness-cost trade-off.

### 5.1.3. COMPARING PENALIZED OT AND COST LEARNING

Figure 3 b. compares our two relaxed approaches across our two datasets, allowing us to highlight the effect of data geometry. For the Gaussian dataset, the cost-fairness trade-off of penalized OT, cost learning with a Mahalanobis metric,

and cost learning with an MLP are similar. On the Circles dataset, however, the penalized approach achieves a superior trade-off, reaching any level of fairness at lower cost than the cost learning methods. This is expected, as penalized OT imposes no restrictions on the transport plans (see Section 4.3). Moreover, cost learning with the Mahalanobis parameterization cannot reduce fairness loss below $10^{-2}$, whereas the MLP variant is flexible enough to transform the data nonlinearly, which is required to closely match the fairness target on this problem. Finally, for both datasets, increasing the entropic penalty $\varepsilon$ in vanilla OT fails to achieve the desired fairness, highlighting the relevance and effectiveness of our approaches.

### 5.1.4. ASSESSING THE REUSABILITY OF LEARNED COSTS

A key advantage of our cost learning approach is that, once the cost function is learned, it can be applied to match new samples using vanilla OT without retraining. In contrast, the penalized optimal transport plan cannot be reused, requiring a new penalized OT problem to be solved for each incoming batch of data.

In this section, we quantify the benefits of reusing the learned cost on new samples in terms of *inference time* and *fairness generalization*. To this end, we first learn a Mahalanobis cost and an MLP-based cost on the Gaussian problem, using a training dataset of 1000 samples from $X$ and 100 samples from $Y$. We subsequently draw test datasets of 500 new samples from $X$ and 50 new samples from $Y$, solve the vanilla entropic OT problem using the learned costs, and record both the inference time and fairness loss of the resulting transport plan for each test dataset. For comparison, we also report the inference time of penalized OT and the fairness of vanilla OT on the same test data. Results are shown in Figure 4.

We observe that using the learned cost allows matching new samples with significantly lower inference time than the penalized approach since the latter requires solving the full penalized optimal transport problem each time new samples are added to the dataset. Furthermore, fairness levels on new samples remain close to those achieved during training. While a generalization gap is present—larger for the MLP cost than for the Mahalanobis cost, as expected—this demonstrates that the learned costs generalize to new samples and achieve substantially better fairness than vanilla OT.

### 5.2. Semi-Synthetic Dating App Experiment

We conduct further experiments on a semi-synthetic dating app dataset obtained from Kaggle.[4] To enhance the realism of the dataset, we subsample it so that the joint

---

[4]Kaggle Dating App Behavior Dataset

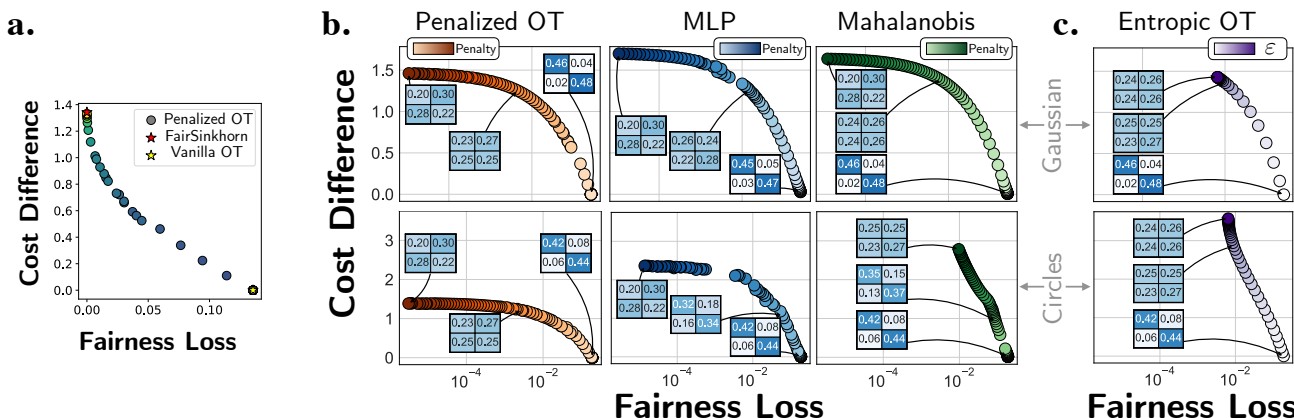

*Figure 3.* **a.** `FairSinkhorn` achieves perfect fairness with a high transport cost while `Sinkhorn` achieves low transport cost with low fairness. The penalized OT interpolates between `Sinkhorn` and `FairSinkhorn`. **b.** Cost-fairness trade-off of our penalized and cost-learning approaches on both datasets for varying fairness penalties. **c.** Vanilla entropic OT with different values of $\varepsilon$ is included as a baseline. We also display the fairness targets reached at a few selected points.

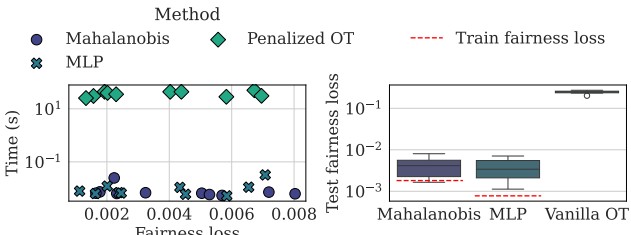

*Figure 4.* **Left:** Inference time of penalized OT and cost-learning approaches, highlighting the much faster inference of the cost-learning methods once the cost function is learned. **Right:** Fairness levels achieved on new samples using the learned cost function, with vanilla OT shown as a baseline. Leveraging the learned cost allows new samples to attain fairness comparable to that observed during training.

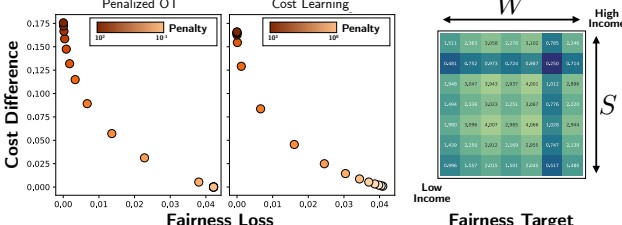

*Figure 5.* Comparison of penalized OT (**left**) and cost learning (**center**) on the semi-synthetic dating app dataset with the fairness target specified on the **right**.

distribution of gender, education level and income matches US demographics. We construct a feasible matching matrix based on reported sexual orientation (SO), excluding some gender-SO combinations for which we lacked sufficient knowledge to reliably determine potential matches. All details on pre-processing can be found in Appendix D.3. In this experiment, the sensitive attribute is income, divided into seven levels ranging from very low to very high. The fairness target is defined as follows: the $(i, j)$ entry is given by $\mathbb{P}(S = s_i) \times \mathbb{P}(W = w_j)$, where $S$ and $W$ are random variables representing the sensitive attribute of individuals from $\mathcal{X}$ and $\mathcal{Y}$, taking values in $\{s_1, \ldots, s_7\}$ and $\{w_1, \ldots, w_7\}$, respectively. The resulting fairness target and associated cost-fairness trade-offs are shown in Figure 5. This experiment extends our previous findings from the synthetic setting (two groups, two-dimensional data) to a high-dimensional setting with multiple groups. In particular, we observe the same convex-shaped trade-off curve, together with nearly identical performance for both methods.

## 6. Conclusion

We introduced a novel notion of group fairness for optimal transport and propose an algorithm that computes entropic regularized transport with perfect fairness. To improve the cost–fairness trade-off, we developed two relaxed approaches: (i) a convex penalization of the entropic OT objective, and (ii) a cost-learning-based method. Both approaches are supported by theoretical results. We hope our work will foster further research on fairness constraints in optimal transport.

**Limitations & future work.** Our work is limited to group fairness, and does not readily extend to alternative notions of fairness such as individual fairness. We believe this is a valuable direction for future work. While our theoretical contributions characterize the sample complexity and deviation probability of our approaches, a throughout analysis of the fairness-bias trade-off is still an open question. Lastly, it would also be interesting to model and evaluate the effects of a matching on downstream individual outcome (i.e. how does a matching of students to universities affect the expected salary for students).

**Acknowledgments.** This work was partially conducted while LB was a member of Inria Montpellier. LB's work at EPFL is supported by an EPFL AI Center Postdoctoral Fellowship. Parts of the experiments for this article were run on the RCP cluster at EPFL. LB thanks Quentin Berthet and Hugo Subtil for insightful discussions. FA's work was partially supported by a French government grant managed by the Agence Nationale de la Recherche under the France 2030 program, reference SMATCH ANR-22-PESN-0003. LB and AB's work was partially supported by a French government grant managed by the Agence Nationale de la Recherche under the France 2030 program, reference SSF-ML-DH ANR-22-PESN-0014. We thank the anonymous reviewer and all participants of the EurIPS workshop *Unifying Perspectives on Learning Biases* whose comments helped improving our work, and the anonymous ICML reviewers.

## Impact Statement

This paper presents a novel way to enforce fairness constraints in OT-based matchings. Matchings are omnipresent in modern societies and govern many aspects of individual lives, such as romantic encounters, employment, housing and medical treatment. Our fairness criterion allows regulators and market designers to enforce fairness targets in a principled manner. We see two main possible impacts of our work. First, we hope that our work will help decision-makers implement more fair matching mechanisms. Secondly, our definition of fairness in matchings might be used as a criterion to analyze existing matching mechanisms and audit them. We stress that the design of the fairness target remains a political, philosophical and ethical decision; our results do not prescribe how such targets should be defined. We also highlight that OT serves as an abstraction for matching problems in our setup, and that it does not fully convey the complexity of real-world matchmaking.

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

# A. Extended Related Work

## A.1. Fairness in Matching Mechanisms

Since the pioneering work of Gale & Shapley (1962) on the stable matching problem, a series of works have provided theoretical studies of the fairness in matching mechanisms. A first line of research focuses on the stability of GS-like algorithms under constraints akin to individual fairness, ensuring that similar individuals receive similar outcomes by the decision-making process. For instance, Karni et al. (2022) analyze the Gale-Shapley (GS) algorithm through the lens of *preference informed individual fairness*. This criterion allows for small deviations from individual fairness, which requires that similar individuals should be treated similarly. They devise a fair variant of the GS algorithm that ensure both approximately fair and stable matchings. Similarly, Devic et al. (2023) analyze merit-based fairness of one-to-one matching mechanisms, a notion close to individual fairness, under partial uncertainty over individual's merit. A second line of research introduces metrics of fairness closer to group fairness, with a specific focus on kidney paired donation programs, a singular matching problem subject to domain specific constraints — in this setting, donors and patients must be matched by forming cycles due to legal constraints, which leads to a reformulation of the matching problem as a constrained integer programming problem. In their seminal work, Ashlagi & Roth (2014) unveil an unfair market collapse phenomenon in kidney pair donation programs: instead of introducing all patients to the matching system, hospitals internally operate easy matches i.e. do not participate fully in the exchange mechanism. This leads to a loss in transplantation opportunities for highly-sensitized patients that are hard to match. Dickerson et al. (2014) and following work by Zhang et al. (2026) also consider the case of kidney paired donations programs. Their definition of fairness enforces a minimal amount of program participation for the group of highly-sensitized patients. More recently, Lobo et al. (2025) consider a combinatorial multimatching problem under uncertainty: in their setup, the goal is to match agents and resources without exact knowledge of their pairwise valuations. Their definition of fairness relies on the average utility within predefined groups, which is closest to our definition among previous work.

Our contribution differentiates from these works in two aspects. First, we do not consider iterative matching mechanisms that operate at the individual level such as the GS algorithm: our work focuses on matching through (entropic) optimal transport. Secondly, our definition of fairness is different from previously introduced definitions, and highly modular to accommodate fairness preferences of a central planner. Instead of requiring individual fairness, building on individual rankings over possible assignments or participation constraints, we consider a matching problem between two populations (f.i. students and universities) further structured by a sensitive attribute (f.i. privileged and unprivileged students, and elite and non-elite universities). We enforce matching quotas given by a central planner, that specify for each subgroup the proportion of individuals that should be matched to subgroups in the second population.

## A.2. Fairness in Bipartite Graph Matching

A closely related topic to our working is bipartite matching in graphs (Godsil, 1981; Zdeborová & Mézard, 2006; Noiry et al., 2021). Given a graph $(\mathcal{U} \cup \mathcal{V}, \mathcal{E})$ made of two sets of vertices $\mathcal{U}$ and $\mathcal{V}$ and edges $\mathcal{E} = (e_{ij})_{i \in \mathcal{U}, j \in \mathcal{V}}$, the goal is to select a one-to-one matching between vertices in $\mathcal{U}$ and vertices in $\mathcal{V}$, under the constraint that two vertices can only be matched if they are linked by an edge in $\mathcal{E}$. Some works have considered fairness constraints in this setup. For instance, Castera et al. (2025) consider this problem with ad-hoc partitions of one side of the graph in disjoint groups, and study the maximal number of individuals that may be matched per group. This is similar to definitions of fairness through participation ratios discussed earlier in the work of Zhang et al. (2026). Similarly, Panda et al. (2024) study a doubly constrained setup in which the set of items $\mathcal{U}$ is structured through groups. They enforce both individual-fairness and group-fairness inspired constraints. In their work, $(a)$ individuals in $\mathcal{U}$ have preferences over individuals in $\mathcal{V}$, and $b)$ every individual in $\mathcal{V}$ can only accept an certain number of individuals from each subgroup in $\mathcal{U}$.

Bipartite matching is hence fundamentally different from our setting and somewhat closer to the literature on fair division (Steinhaus, 1949; Weller, 1985; Budish, 2011) since $a)$ we allow for mass splitting between individuals on both sides ; $b)$ individuals are characterized by features in our setting, and not by their relative position within a graph $\mathcal{E}$ ; $c)$ our problem is symmetric in the sens that the two matched measures play the same role.

## A.3. Fairness and Optimal Transport

Recent work has drawn many fruitful connection between fairness and optimal transport, primarily by using OT as a tool to obtain or characterize fair learning algorithms (Gordaliza et al., 2019; Gouic et al., 2020; Chzhen et al., 2020; Chiappa et al.,

2020; Buyl & Bie, 2022; Hu et al., 2023; Xian et al., 2023; Chowdhary et al., 2024; Xiong et al., 2024; Divol & Gaucher, 2024). A central insight of this line of work is that learning an optimal fair predictor can be formulated as a Wasserstein barycenter problem (Gouic et al., 2020; Chzhen et al., 2020; Divol & Gaucher, 2024). This perspective is orthogonal to ours: OT serves as an auxiliary mechanism for analyzing and enforcing fairness in downstream prediction tasks, whereas our goal is to study and enforce fairness of the optimal transport itself.

Closer in spirit to our work is the recent article by Nguyen et al. (2025), which imposes fairness constraints directly on a Wasserstein barycenter by requiring approximate equality of sliced-Wasserstein distances **SW** (Nadjahi et al., 2020) to multiple marginals $\mu_1, \ldots, \mu_K$.

$$\min_{\mu} \sum_{k=1}^{K} \mathbf{SW}(\mu, \mu_k) \text{ s.t } \sum_{i=1}^{K} \sum_{j=1}^{K} \left| \mathbf{SW}(\mu, \mu_i) - \mathbf{SW}(\mu, \mu_j) \right| \leq \varepsilon.$$

where $\mathbf{SW}(\cdot, \cdot)$ is the sliced-Wasserstein distance between distributions (Nadjahi et al., 2020) and $\varepsilon > 0$ is a tolerance threshold. Similarly to their work, we enforce additional constraints on the optimal transport to match it to a definition of fairness based on subgroups. However, we study the problem of computing transport plans that satisfy explicit mass constraints between groups defined by sensitive attributes, a setting that, to the best of our knowledge, has not been previously considered.

### A.4. Constrained and Statistical Optimal Transport

Our work borrows from recent developments in constrained optimal transport, which seeks to enforce structural properties on the transport plan (Courty et al., 2016; Blondel et al., 2018; Paty & Cuturi, 2019; Liu et al., 2023); for instance, Korman & McCann (2015) analyze a variant of optimal transport in which the amount of mass that can be transported between two units is upper bounded. Specifically, we build on Rakotomamonjy et al. (2015), Genevay et al. (2019) and Rigollet & Stromme (2025) to derive finite sample guarantees for a penalized optimal transport problem, and to bound the deviations of fairness with learned cost functions.

## B. A Modified Sinkhorn Algorithm for Fair Optimal Transport

The following is an immediate consequence of first order conditions on the Lagrangian of Problem 3.

**Proposition B.1.** There exists $\mathbf{f} \in \mathbb{R}^n, \mathbf{g} \in \mathbb{R}^m$ and $\mathbf{h} = [h_{sw}]_{sw} \in \mathbb{R}^{K_s \times K_w}$ such that the solution to Problem 3 has the form

$$\mathbf{\Pi} = \mathrm{diag}\big(e^{\mathbf{f}/\varepsilon}\big) \big(\mathbf{K} \odot \mathbf{H}\big) \mathrm{diag}\big(e^{\mathbf{g}/\varepsilon}\big)$$

where

$$\mathbf{K} := \left[ e^{-\mathbf{C}_{ij}/\varepsilon} \right]_{ij} \text{ and } \mathbf{H} := \sum_{sw} e^{h_{sw}/\varepsilon} \mathbf{B}_{sw}.$$

*Proof.* Introducing dual variables $\mathbf{f} \in \mathbb{R}^n, \mathbf{g} \in \mathbb{R}^m$ and $\mathbf{H} = (h_{sw})_{sw} \in \mathbb{R}^{K_s \times K_w}$, the Lagrangian writes

$$\mathcal{E}\big(\mathbf{\Pi}, \mathbf{f}, \mathbf{g}, \mathbf{h}\big) = \mathrm{Tr}[\mathbf{\Pi}^\top \mathbf{C}] + \varepsilon \mathbf{KL}(\mathbf{\Pi}) - \mathbf{f}^\top(\mathbf{\Pi} \mathbb{1}_m - \mathbf{a}) - \mathbf{g}^\top(\mathbf{\Pi}^\top \mathbb{1}_n - \mathbf{b})$$
$$- \sum_{s,w} h_{sw} \Big[ \mathrm{Tr}\big[\mathbf{\Pi}^\top \mathbf{B}_{sw}\big] - \mathbf{F}_{sw} \Big].$$

First order conditions yield for every $i, j$

$$\frac{\partial \mathcal{E}\big(\mathbf{\Pi}, \mathbf{f}, \mathbf{g}, \mathbf{h}\big)}{\partial \mathbf{\Pi}_{ij}} = \mathbf{C}_{ij} + \varepsilon \log\big(\mathbf{\Pi}_{ij}\big) + \varepsilon - \mathbf{f}_i - \mathbf{g}_j - \sum_{sw} h_{sw} \big[\mathbf{B}_{sw}\big]_{ij} = 0$$

which we may rewrite as

$$\mathbf{\Pi}_{ij} = \exp\left(\mathbf{f}_i/\varepsilon\right) \exp\left(-\varepsilon^{-1}\mathbf{C}_{ij} + \varepsilon^{-1}\sum_{sw} h_{sw}\left[\mathbf{B}_{sw}\right]_{ij} - 1\right) \exp\left(\mathbf{g}_j/\varepsilon\right)$$

$$= \exp\left(\mathbf{f}_i/\varepsilon\right) \exp(-\mathbf{C}_{ij}/\varepsilon - 1) \prod_{s'=1}^{K_s} \prod_{w'=1}^{K_w} \exp\left(h_{s'w'}[\mathbf{B}_{s'w'}]_{ij}/\varepsilon\right) \exp\left(\mathbf{g}_j/\varepsilon\right) .$$

Now, remark that for any $(i,j) \in \{1,\dots,n\}^2$, there is only one $(s,w) \in \{1,\dots,K_s\} \times \{1,\dots,K_w\}$ such that $[\mathbf{B}_{sw}]_{i,j} \neq 0$. Therefore, in the product $\prod_{s'=1}^{K_s} \prod_{w'=1}^{K_w} \exp\left(h_{s'w'}[\mathbf{B}_{s'w'}]_{ij}\right)$, there is only one term distinct from 1. As a consequence

$$\prod_{s'=1}^{K_s} \prod_{w'=1}^{K_w} \exp\left(h_{s'w'}[\mathbf{B}_{s'w'}]_{ij}/\varepsilon\right) = \prod_{s'=1}^{K_s} \prod_{w'=1}^{K_w} \exp\left(h_{s'w'}\mathbb{1}_{\mathbf{s}_i=s'}\mathbb{1}_{\mathbf{w}_j=w'}/\varepsilon\right)$$

$$= \sum_{s'=1}^{K_s} \sum_{w'=1}^{K_w} \exp\left(h_{s'w'}/\varepsilon\right)\mathbb{1}_{\mathbf{s}_i=s'}\mathbb{1}_{\mathbf{w}_j=w'}$$

$$= \sum_{s'=1}^{K_s} \sum_{w'=1}^{K_w} \exp\left(h_{s'w'}/\varepsilon\right)\left[\mathbf{B}_{s'w'}\right]_{ij} .$$

This leads to the matrix form

$$\mathbf{\Pi} = \mathrm{diag}\left(e^{\mathbf{f}/\varepsilon}\right)\left(\mathbf{K} \odot \mathbf{H}\right)\mathrm{diag}\left(e^{\mathbf{g}/\varepsilon}\right)$$

where

$$\mathbf{K} := e^{-\mathbf{C}/\varepsilon - 1}$$

$$\mathbf{H} := \sum_{s=1}^{K_s} \sum_{w=1}^{K_w} \exp(h_{sw}/\varepsilon) \odot \mathbf{B}_{sw}.$$

$\square$

# C. Proofs

### C.1. Proof of Proposition 3.2

**Lemma C.1.** Let $\mu \in \mathcal{P}(\mathcal{X} \times \mathcal{S})$ and $\eta \in \mathcal{P}(\mathcal{Y} \times \mathcal{W})$. Let $p \in \mathcal{P}(\mathcal{S})$ and $q \in \mathcal{P}(\mathcal{W})$ be obtained from $\mu$ and $\eta$ by marginalizing, respectively, $x$ and $y$, that is,

$$p(S = s) = \mu(\mathcal{X} \times \{s\}) \tag{8}$$
$$q(W = w) = \eta(\mathcal{Y} \times \{w\}). \tag{9}$$

Finally let $F \in \Pi(p,q)$.

There exists $\pi \in \Pi(\mu,\eta)$ such that $\pi$ is $F-$fair, that is

$$\pi\left(\mathcal{X} \times \{s\} \times \mathcal{Y} \times \{w\}\right) = F(S = s, W = w).$$

*Proof.* Given measurable sets $A \subseteq \mathcal{X}$ and $B \subseteq \mathcal{Y}$ let $\pi$ be defined as

$$\pi\left(A \times \{s\} \times B \times \{w\}\right) := \frac{F(S = s, W = w)\mu(A \times \{s\})\eta(B \times \{w\})}{p(S = s)q(W = w)}$$

and let's check that **i)** $\pi \in \Pi(\mu,\eta)$ and **ii)** $\pi$ is $F-$fair:

i) We want to check that $\pi(A \times \{s\} \times \mathcal{Y} \times \mathcal{W}) = \mu(A \times \{s\})$ and similarly for the other marginal. To this end note that

$$\pi(A \times \{s\} \times \mathcal{Y} \times \mathcal{W}) = \sum_{w \in \mathcal{W}} \frac{F(S = s, W = w)\mu(A \times \{s\})\eta(\mathcal{Y} \times \{w\})}{p(S = s)q(W = w)}$$

$$= \mu(A \times \{s\}) \sum_{w \in \mathcal{W}} \frac{F(S = s, W = w)}{p(S = s)},$$

where the second equality follows from (9). To finish note that from $F \in \Pi(p, q)$ follows that $\sum_{w \in \mathcal{W}} F(S = s, W = w) = p(S = s)$. A similar argument shows that $\pi(\mathcal{X} \times \mathcal{S} \times B \times \{w\}) = \eta(B \times \{w\})$.

ii) To see that $\pi$ is $F-$fair note that it is immediate from (8) and (9) that

$$\pi(\mathcal{X} \times \{s\} \times \mathcal{Y} \times \{w\}) = \frac{F(S = s, W = w)\mu(\mathcal{X} \times \{s\})\eta(\mathcal{Y} \times \{w\})}{p(S = s)q(W = w)}$$

$$= F(S = s, W = w).$$

$\square$

**Proposition C.1.** *Assume that $\mu$ and $\eta$ have bounded support. Let $F$ be a coupling of the marginals of $\mu$ and $\eta$ denoted by $p$ and $q$ defined via (8)-(9). Then there exists a unique $F$-fair transport plan.*

*Proof.* Assume $\mathcal{X}$ and $\mathcal{Y}$ to be compact. Similar to the proof of Theorem 1.4. in (Santambrogio, 2015) we can prove that

$$\Lambda := \{\pi \in \mathcal{P}(\mathcal{X} \times \{0, 1\} \times \mathcal{Y} \times \{0, 1\}) : \pi_{SW} = F\}$$

is compact with respect to the weak topology: Let $\pi_n$ be a sequence in $\Lambda$. They are probability measures, so that their mass is 1, and hence they are bounded in the dual of $C(\mathcal{X} \times \{0, 1\} \times \mathcal{Y} \times \{0, 1\})$. Hence usual weak-$\star$ compactness in dual spaces guarantees the existence of a subsequence $\pi_n \rightharpoonup \pi$ converging to a probability $\pi$. We just need to check $\pi \in \Lambda$. This may be done by fixing $\phi \in C(\{0, 1\} \times \{0, 1\})$ and from $\pi_n \in \Lambda$ it follows that

$$\int \phi \, d\pi_n = \int \phi \, d[(\pi_n)_{SW}] = \int \phi \, dF = \sum_{s,w} \phi(s, w) F(S = s, W = w).$$

Now pass to the limit to obtain

$$\int \phi \, d\pi_{SW} = \int \phi \, d\pi = \sum_{s,w} \phi(s, w) F(S = s, W = W)$$

which shows that $\pi \in \Lambda$.

To finish, just note that $\Pi_{\text{fair}}^{\mathbf{F}} = \Pi(\mu, \eta) \cap \Lambda$ is the intersection of two compact sets (with respect to weak topology) and Lemma C.1 establishes that it is non-empty. Continuity of the map defining the transport problem is enough to conclude the existence of a minimizer. Uniqueness is then a consequence of the strict convexity of KL. $\square$

### C.2. Proof of Theorem 4.2

**Notation.** Let $\mathbf{F} \in \mathbb{R}^{K_s \times K_w}$ a given a fairness target. We adopt the following notational conventions, $\mathbf{u} \in [0, 1]^{K_s}$, $\mathbf{u}^\top \mathbf{1}_{K_s} = 1$, and $\mathbf{t} \in [0, 1]^{K_w}$, $\mathbf{t}^\top \mathbf{1}_{K_w} = 1$, we define the penalization to be

$$\xi_{sw}(\mathbf{u}, \mathbf{t}) := \mathbf{u}_s \mathbf{t}_w - \mathbf{F}_{sw}, \tag{10}$$

so that given a measure $\pi \in \mathcal{M}(\mathcal{X} \times [K_s] \times \mathcal{Y} \times [K_w])$, define

$$\mathcal{L}_{\mathbf{F}}(\pi) := \sum_{s,w \in [K_s] \times [K_w]} \langle \xi_{sw}, \pi \rangle^2 = \sum_{s,w \in [K_s] \times [K_w]} \left( \int \xi_{sw}(z, w) \, d\pi(x, z, y, w) \right)^2,$$

where we use the duality pairing notation and integral notation interchangeably. For any distribution $\mu, \eta$ over a set $\mathcal{X}$ such that $\mu$ is absolutely continuous with respect to $\eta$ (i.e., $\mu \ll \eta$), we denote by $\frac{d\mu}{d\eta}$ the Radon-Nikodym derivative of $\mu$ with respect to $\eta$, and

$$\mathbf{KL}(\mu||\eta) := \int_{\mathcal{X}} \log\left(\frac{d\mu}{d\eta}(x)\right) d\mu(x).$$

Given any two measures $\mu \in \mathcal{M}(\mathcal{X} \times [K_s])$ and $\eta \in \mathcal{M}(\mathcal{Y} \times [K_w])$ define

$$m^\star(\mu, \eta) := \min_{\pi \in \Pi(\mu, \eta)} \langle c, \pi \rangle + \varepsilon \mathbf{KL}(\pi||\mu \otimes \eta) + \lambda \mathcal{L}_{\mathbf{F}}(\pi). \tag{11}$$

The goal is to prove the following result

**Theorem 4.2.** Let $\mathcal{X}$ and $\mathcal{Y}$ be compact subsets of $\mathbb{R}^d$ and let $\mu$ and $\eta$ be probability measures on $\mathcal{X} \times [K_s]$ and $\mathcal{Y} \times [K_w]$, respectively. Let $(x_i, s_i)_{i=1}^n$ be $n$ independent and identically distributed (i.i.d.) samples from $\mu$ and let $(y_j, w_j)_{j=1}^n$ be $n$ i.i.d. samples from $\eta$; assume further that the samples from $\mu$ are independent from those from $\eta$. Finally, suppose that the cost $c$ is $L$-Lipschitz continuous and infinitely differentiable. Then

$$\mathbb{E}_{\mu \otimes \eta}|m^\star(\mu_n, \eta_n) - m^\star(\mu, \eta)| \leq \mathcal{O}(1/\sqrt{n}^{-1}), \tag{12}$$

*Proof.* Let $m_\infty^\star := m^\star(\mu, \eta)$ and $m_n^\star := m^\star(\mu_n, \eta_n)$, and $\pi_\infty^\star$ be the minimizer attaining the $m_\infty^\star$ that is a measure on $\mathcal{X} \times [K_s] \times \mathcal{Y} \times [K_w]$ with marginals $\mu$ and $\eta$ such that

$$m_\infty^\star = \langle c, \pi_\infty^\star \rangle + \varepsilon \mathbf{KL}(\pi_\infty^\star||\mu \otimes \eta) + \lambda \mathcal{L}_{\mathbf{F}}(\pi_\infty^\star).$$

Let $p_\infty^\star$ denote the Radon–Nikodym density of $\pi_\infty^\star$ with respect to $\mu \otimes \eta$ (note that the Kullback–Leibler term forces $\pi_\infty^\star$ to be absolutely continuous with respect to $\mu \otimes \eta$).

The idea is to "sandwich" the random variable $m_n^\star - m_\infty^\star$ between two random variables with expectation upper bounded by $\mathcal{O}(1/\sqrt{n})$, that is, to find random variables $Y_n, Z_n$ such that

$$Y_n - m_\infty^\star \leq m_n^\star - m_\infty^\star \leq Z_n - m_\infty^\star \tag{13}$$

and

$$\mathbb{E}[|Y_n - m_\infty^\star|] \leq \mathcal{O}(1/\sqrt{n}^{-1}) \quad \text{and} \quad \mathbb{E}[|Z_n - m_\infty^\star|] \leq \mathcal{O}(1/\sqrt{n}^{-1}).$$

To see that this is enough to establish (12), observe that (13) implies that

$$|m_n^\star - m_\infty^\star| \leq \max(|Y_n - m_\infty^\star|, |Z_n - m_\infty^\star|) \leq |Y_n - m_\infty^\star| + |Z_n - m_\infty^\star|.$$

**Lower bound.** To define $Y_n$ start by observing that by linearizing the fairness penalization, we can incorporate this linearization into the transport cost, so that the problem is now a proper entropic optimal transport one. This allows us to leverage existing results on the sample complexity of optimal transport. Moreover, the convexity of $\mathcal{L}_{\mathbf{F}}$ implies that this linearization does yield a lower bound. To this end, note that for any measure $\pi \in \mathcal{M}(\mathcal{X} \times [K_s] \times \mathcal{Y} \times [K_w])$ and any $s, w \in [K_s] \times [K_w]$, the inequality $0 \leq (\langle \xi_{sw}, \pi - \pi_\infty^\star \rangle)^2 = (\langle \xi_{sw}, \pi \rangle)^2 + (\langle \xi_{sw}, \pi_\infty^\star \rangle)^2 - 2\langle \xi_{sw}, \pi \rangle \langle \xi_{sw}, \pi_\infty^\star \rangle$, implies that

$$\mathcal{L}_{\mathbf{F}}(\pi) \geq -\mathcal{L}_{\mathbf{F}}(\pi_\infty^\star) + 2 \sum_{(s,w) \in [K_s] \times [K_w]} \langle \langle \xi_{sw}, \pi_\infty^\star \rangle \xi_{sw}, \pi \rangle$$

$$= \mathcal{L}_{\mathbf{F}}(\pi_\infty^\star) + \langle 2 \sum_{s,w \in [K_s] \times [K_w]} \langle \xi_{sw}, \pi_\infty^\star \rangle \xi_{sw}, \pi - \pi_\infty^\star \rangle.$$

This leads to a lower bound on $m_n^\star$ given by

$$m_n^\star \geq -\lambda \mathcal{L}_{\mathbf{F}}(\pi_\infty^\star) + \underbrace{\min_{\substack{\pi_1 = \mu_n \\ \pi_2 = \eta_n}} \langle \overbrace{c + 2\lambda \sum_{s,w \in [K_s] \times [K_w]} \langle \xi_{sw}, \pi_\infty^\star \rangle \xi_{sw}}^{:=\hat{c}}, \pi \rangle + \varepsilon \mathbf{KL}(\pi||\mu_n \otimes \eta_n)}_{:=\hat{m}_n^\star} := Y_n \tag{14}$$

where $\widehat{m}_n^\star$ is an entropic optimal transport problem with cost $\widehat{c}$. From the sample complexity of optimal transport (see Theorem 18 in (Genevay, 2019)) it follows that

$$\mathbb{E}_{\mu\otimes\eta}\big|\widehat{m}_n^\star - \widehat{m}_\infty^\star\big| \leq \mathcal{O}(\sqrt{n}^{-1}), \tag{15}$$

with $\widehat{m}_\infty^\star$ being the population version of $\widehat{m}_n^\star$, i.e.,

$$\widehat{m}_\infty^\star := \min_{\substack{\pi_1=\mu \\ \pi_2=\eta}} \langle c + 2\lambda \sum_{s,w\in[K_s]\times[K_w]} \langle\xi_{sw}, \pi_\infty^\star\rangle\xi_{sw}, \pi\rangle + \varepsilon\mathbf{KL}(\pi\|\mu\otimes\eta).$$

To finish the lower bound, note that $\mathbb{E}[|Y_n - m_\infty^\star|] \leq \mathcal{O}(\sqrt{n}^{-1})$ follows from (15) provided we show that $m_\infty^\star = \widehat{m}_\infty^\star - \lambda\mathcal{L}_{\mathbf{F}}(\pi_\infty^\star)$; this is the content of Lemma C.2 that can be found in Appendix C.2.1.

**Upper bound.** Let $\hat{\pi}_n^\star$ be a minimizer attaining $\hat{m}_n^\star$ (see (14) for a definition of $\hat{m}_n^\star$). From the fact that $\hat{\pi}_n^\star$ satisfies the marginal constraints defining $m_n^\star$ follows that

$$\min_{\pi\in\Pi(\mu_n,\eta_n)} \langle c,\pi\rangle + \varepsilon\mathbf{KL}\big(\pi\|\mu_n\otimes\eta_n\big) + \lambda\mathcal{L}_{\mathbf{F}}(\pi) = m_n^\star \leq \langle c,\hat{\pi}_n^\star\rangle + \varepsilon\mathbf{KL}\big(\hat{\pi}_n^\star\|\mu_n\otimes\eta_n\big) + \lambda\mathcal{L}_{\mathbf{F}}(\hat{\pi}_n^\star) := Z_n.$$

Let $\hat{c}$ be as in (14) and note that $Z_n$ decomposes as

$$Z_n = \underbrace{\langle\hat{c},\hat{\pi}_n^\star\rangle + \varepsilon\mathbf{KL}\big(\hat{\pi}_n^\star\|\mu_n\otimes\eta_n\big)}_{:=Z_{n_1}} + \underbrace{\lambda\mathcal{L}_{\mathbf{F}}(\hat{\pi}_n^\star) + \langle c-\hat{c},\hat{\pi}_n^\star\rangle}_{:=Z_{n_2}}.$$

Moreover, by definition of $\hat{\pi}_n^\star$ and $\hat{c}$,

$$Z_{n_1} = \widehat{m}_n^\star \quad\text{and}\quad Z_{n_2} = \lambda\mathcal{L}_{\mathbf{F}}(\hat{\pi}_n^\star) - 2\lambda \sum_{(s,w)\in[K_s]\times[K_w]} \big\langle\langle\xi_{sw},\pi_\infty^\star\rangle\xi_{sw}, \hat{\pi}_n^\star\big\rangle.$$

Lemma C.2 shows that $m_\infty^\star = \widehat{m}_\infty^\star - \lambda\mathcal{L}_F(\pi_\infty^\star)$ and, similar to the lower bound part of the proof (see (15)), the sample complexity of entropic optimal transport implies that,

$$\mathbb{E}\big[\big|Z_{n_1} - \widehat{m}_\infty^\star\big|\big] = \mathbb{E}\big[\big|\widehat{m}_n^\star - \widehat{m}_\infty^\star\big|\big] \leq \mathcal{O}(\sqrt{n}^{-1}).$$

Consequently, to establish that $\mathbb{E}[|Z_n - m_\infty^\star|] \leq \mathcal{O}(1/\sqrt{n})$ it is enough to show that

$$\mathbb{E}\big[\big|Z_{n_2} - \big(-\lambda\mathcal{L}_F(\pi_\infty^\star)\big)\big|\big] \leq \mathcal{O}(1/\sqrt{n}). \tag{16}$$

To this end, note that

$$-\lambda\mathcal{L}_{\mathbf{F}}(\pi_\infty^\star) = \lambda\mathcal{L}_{\mathbf{F}}(\pi_\infty^\star) - 2\lambda \sum_{(s,w)\in[K_s]\times[K_w]} \big\langle\langle\xi_{sw},\pi_\infty^\star\rangle\xi_{sw}, \pi_\infty^\star\big\rangle,$$

because, for any $(s,w)\in\{0,1\}^2$, $\big\langle\langle\xi_{sw},\pi_\infty^\star\rangle\xi_{sw}, \pi_\infty^\star\big\rangle = \langle\xi_{sw},\pi_\infty^\star\rangle^2$, hence

$$\mathcal{L}_{\mathbf{F}}(\pi_\infty^\star) = \sum_{(s,w)\in[K_s]\times[K_w]} \big\langle\langle\xi_{sw},\pi_\infty^\star\rangle\xi_{sw}, \pi_\infty^\star\big\rangle.$$

This, in turn, reduces the proof of (16) to showing

$$\mathbb{E}[|\lambda\mathcal{L}_{\mathbf{F}}(\hat{\pi}_n^\star) - \lambda\mathcal{L}_{\mathbf{F}}(\pi_\infty^\star)|] \leq \mathcal{O}(1/\sqrt{n}) \tag{17}$$

and

$$\mathbb{E}\Big[2\lambda\big|\sum_{(s,w)\in[K_s]\times[K_w]} \big\langle\langle\xi_{sw},\pi_\infty^\star\rangle\xi_{sw}, \pi_\infty^\star - \hat{\pi}_n^\star\big\rangle\big|\Big] \leq \mathcal{O}(1/\sqrt{n}). \tag{18}$$

To obtain these two inequalities and complete the proof, it suffices to note that the conclusion of Lemma C.2 allows us to apply Lemma C.4 (sub-Gaussian concentration, which is stronger, readily implies the expectation bound used here), which yields the desired inequality (17). To obtain inequality (18) just note that, for $s,w\in[K_s]\times[K_w]$,

$$\langle\xi_{sw},\pi_\infty^\star\rangle^2 - \langle\xi_{sw},\hat{\pi}_n^\star\rangle^2 = \langle\xi_{sw},\pi_\infty^\star - \hat{\pi}_n^\star\rangle\langle\xi_{sw},\pi_\infty^\star + \hat{\pi}_n^\star\rangle$$

which implies that

$$\left|\langle\xi_{sw},\pi_{\infty}^{\star}\rangle^2 - \langle\xi_{sw},\hat{\pi}_{n}^{\star}\rangle^2\right| \leq 2\|\xi_{sw}\|_{\infty}\left|\langle\xi_{sw},\pi_{\infty}^{\star}-\hat{\pi}_{n}^{\star}\rangle\right|;$$

and the bound in expectation is once more a matter of invoking Lemma C.4. $\qquad\square$

There are two noteworthy dependencies in the constants of our sample complexity upper bound, namely the entropic regularization $\varepsilon$ and the fairness penalty $\lambda$. Our proof proceeds by reducing the sample complexity bound to that of entropic optimal transport. Specifically, we linearize the penalty loss, which results in a shifted cost function $\hat{c}$ (14). Consequently, since the sample complexity grows exponentially with $\frac{\|\hat{c}\|_{\infty}}{\varepsilon}$ (Genevay, 2019, Theorem 3), the scaling on $\varepsilon$ matches that of entropic optimal transport ($e^{\frac{\|c\|_{\infty}}{\varepsilon}}$), and yields the following scaling on $\lambda$: $e^{\frac{\lambda K_s K_w}{\varepsilon}}$. That is, for reasonable values of penalization, i.e., $\lambda K_s K_w \lesssim \|c\|_{\infty}$, the scaling is of the same order as entropic OT. There is also a factor, for both term, of $(1 + \frac{1}{\varepsilon^{\lfloor d/2\rfloor}})$; but we omit it as the same as entropic OT.

### C.2.1. TECHNICAL LEMMAS

**Lemma C.2.** Let $\pi_{\infty}^{\star}$ be a minimizer of

$$\min_{\substack{\pi_1=\mu\\\pi_2=\eta}} F(\pi) := \langle c,\pi\rangle + \varepsilon\mathbf{KL}\big(\pi\|\mu\otimes\eta\big) + \lambda\mathcal{L}_{\mathbf{F}}(\pi) := m_{\infty}^{\star}. \tag{19}$$

Then $\pi_{\infty}^{\star}$ also minimizes

$$\min_{\substack{\pi_1=\mu\\\pi_2=\eta}} F_L(\pi) := \langle c,\pi\rangle + \varepsilon\mathbf{KL}(\pi\|\mu\otimes\eta) + \lambda\mathcal{L}_{\mathbf{F}}(\pi_{\infty}^{\star}) + \Big\langle 2\lambda\sum_{s,w\in[K_s]\times[K_w]}\langle\xi_{sw},\pi_{\infty}^{\star}\rangle\xi_{sw},\pi-\pi_{\infty}^{\star}\Big\rangle \tag{20}$$

$$= \min_{\substack{\pi_1=\mu\\\pi_2=\eta}}\langle c+2\lambda\sum_{s,w\in[K_s]\times[K_w]}\langle\xi_{sw},\pi_{\infty}^{\star}\rangle\xi_{sw},\pi\rangle + \varepsilon\mathbf{KL}(\pi\|\mu\otimes\eta) - \lambda\mathcal{L}_{\mathbf{F}}(\pi_{\infty}^{\star})$$

$$= \widehat{m}_{\infty}^{\star} - \lambda\mathcal{L}_{\mathbf{F}}(\pi_{\infty}^{\star}).$$

and, consequently, $\widehat{m}_{\infty}^{\star} - \lambda\mathcal{L}_{\mathbf{F}}(\pi_{\infty}^{\star}) = m_{\infty}^{\star}$.

*Proof.* The proof is inspired by the one from (Rakotomamonjy et al., 2015, Proposition 2.1). Assume $\pi_{\infty}^{\star}$ minimizes Equation (19). The presence of entropic regularization implies that $\pi_{\infty}^{\star}\ll\mu\otimes\eta$. Let

$$g_L = 2\sum_{s,w\in[K_s]\times[K_w]}\langle\xi_{sw},\pi_{\infty}^{\star}\rangle\xi_{sw}.$$

We already know that for any $\pi\in\Pi(\mu,\eta)$

$$\mathcal{L}_{\mathbf{F}}(\pi) \geq \mathcal{L}_{\mathbf{F}}(\pi_{\infty}^{\star}) + \langle g_L,\pi-\pi_{\infty}^{\star}\rangle.$$

Define

$$g_{\mathrm{KL}} = \log\left(\frac{d\pi_{\infty}^{\star}}{d(\mu\otimes\eta)}\right),$$

we prove, for any $\pi\in\Pi(\mu,\eta)$,

$$\mathbf{KL}(\pi\|\mu\otimes\eta) \geq \mathbf{KL}(\pi_{\infty}^{\star}\|\mu\otimes\eta) + \langle g_{\mathrm{KL}},\pi-\pi_{\infty}^{\star}\rangle,$$

with convex analysis[5], and thus

$$F(\pi) \geq F(\pi_{\infty}^{\star}) + \langle c+\lambda g_L+\varepsilon g_{\mathrm{KL}},\pi-\pi_{\infty}^{\star}\rangle.$$

---

[5]If $\pi$ is singular with respect to $\mu\otimes\eta$, the inequality is trivial. Assume $\pi\ll\mu\otimes\eta$, we define for $r\in[0,1]$, $\pi_r = (1-r)\pi_{\infty}^{\star}+r\pi\in\Pi(\mu,\eta)$ and $\phi(r) = \mathbf{KL}(\pi_r\|\mu\otimes\eta)$. By convexity of $\phi$, $\phi(r)\geq\phi(0)+r\lim_{h\to0^+}\frac{\phi(h)-\phi(0)}{h}$. We compute the previous directional limit using the monotone convergence theorem as in Csiszár (1975, Lemma 2.1).

Consequently, for any $\pi \in \Pi(\mu, \eta)$ such that $\pi \ll \mu \otimes \eta$[6],

$$\langle c + \lambda g_L + \varepsilon g_{\mathrm{KL}}, \pi - \pi_\infty^\star \rangle \geq 0,$$

otherwise, for $r \in [0, 1]$, $\pi_r = (1 - r)\pi_\infty^\star + r\pi \in \Pi(\mu, \eta) \cap \{\eta; \eta \ll \mu \otimes \eta\}$, we could find $r^\star \in (0, 1]$, such that $F(\pi_{r^\star}) < F(\pi_\infty^\star)$. In fact, by contradiction, assume $\langle c + \lambda g_L + \varepsilon g_{\mathrm{KL}}, \pi - \pi_\infty^\star \rangle < 0$, as $\lim_{r \to 0^+} \frac{F(\pi_r) - F(\pi_\infty^\star)}{r} = \langle c + \lambda g_L + \varepsilon g_{\mathrm{KL}}, \pi - \pi_\infty^\star \rangle$, we have, by continuity, the existence of a sufficiently small $r^\star \in (0, 1]$ such that $F(\pi_{r^\star}) < F(\pi_\infty^\star)$. Moreover, as $F_L(\pi_\infty^\star) = F(\pi_\infty^\star)$, for any $\pi \in \Pi(\mu, \eta)$

$$F_L(\pi) \geq F_L(\pi_\infty^\star) + \langle c + \lambda g_L + \varepsilon g_{\mathrm{KL}}, \pi - \pi_\infty^\star \rangle \geq F_L(\pi_\infty^\star).$$

Hence, $\pi_\infty^\star$ is also a minimizer of Equation (20). □

### C.3. Proof of Theorem 4.4

We begin by showing the following Lemma, which mimics the arguments of Rigollet & Stromme (2025).

**Lemma C.3.** Let $f_\star, g_\star$ be dual potentials such that $\int g_\star(y) d\eta(y) = 0$. Then we have

$$\|f_\star\|_\infty \leq \|c_\theta\|_\infty$$
$$\|g_\star\|_\infty \leq \|c_\theta\|_\infty.$$

*Proof.* Let $f_\star, g_\star$ be two dual potentials. Since

$$\int e^{-\frac{1}{\varepsilon}\left(c_\theta(x,y) - f_\star(x) - g_\star(y)\right)} d\eta(y) = 1,$$

we have, by using the fact that $\eta$ and $\mu$ have bounded support, that

$$1 = \int e^{-\frac{1}{\varepsilon}\left(c_\theta(x,y) - f_\star(x) - g_\star(y)\right)} d\eta(y) \geq e^{-\frac{1}{\varepsilon}\left(\max_{x,y} c_\theta(x,y) - f_\star(x)\right)} \int e^{\frac{1}{\varepsilon}g_\star(y)} d\eta(y).$$

By Jensen's inequality,

$$\int e^{\frac{1}{\varepsilon}g_\star(y)} d\eta(y) \geq e^{\frac{1}{\varepsilon}\int g_\star(y) d\eta(y)} = 1$$

using the convention $\int g_\star(y) d\eta(y) = 0$. We are thus left with

$$e^{-\frac{1}{\varepsilon}\left(\max_{x,y} c_\theta(x,y) - f_\star(x)\right)} \leq 1$$

which can be rearranged to yield

$$f_\star(x) \leq \|c_\theta\|_\infty$$

$\mu$-almost everywhere where $\|c_\theta\|_\infty := \max_{x,y} c(x, y)$. We may now using this inequality to write that

$$f_\star(x) + g_\star(y) - c_\theta(x, y) \leq f_\star(x) + g_\star(y) \leq g_\star(y) + \|c_\theta\|_\infty$$

since $c_\theta(x, y) \geq 0$. This gives

$$e^{-\frac{1}{\varepsilon}\left(c_\theta(x,y) - f_\star(x) - g_\star(y)\right)} \geq e^{-\frac{1}{\varepsilon}\left(g_\star(y) + \|c_\theta\|_\infty\right)}.$$

Using again the fact that the left-hand side of this inequality is equal to 1, we get

$$g_\star(y) \geq -\|c_\theta\|_\infty.$$

---

[6]If instead $\pi$ is singular with respect to $\mu \otimes \eta$, then $F(\pi) = F_L(\pi) = \infty$, so such measures are trivially excluded from being minimizers of either functional.

Leveraging the same argument than Rigollet & Stromme (2025), we have

$$\int f_\star(x) d\mu(x) \geq 0,$$

which in turn allows us to conclude by symmetrical arguments and to get

$$\|f_\star\|_\infty \leq \|c_\theta\|_\infty$$
$$\|g_\star\|_\infty \leq \|c_\theta\|_\infty.$$

$\square$

Using this Lemma, we get the following result which extends Theorem 6 from Rigollet & Stromme (2025) to parametrized costs.

**Lemma C.4.** Let $\varphi \in L^\infty(\mu \otimes \eta)$. For all $t > 0$, with probability at least $1 - 18e^{-t^2}$, we have

$$\left| \int \varphi d(\pi_n - \pi_\star) \right| \lesssim \frac{\exp\left(5R_\Theta \varepsilon^{-1}\right) \|\varphi\|_{L^\infty(\mu \otimes \eta)} \cdot t}{\sqrt{n}}.$$

*Proof.* We may now write

$$p_\star(x,y) = e^{-\frac{\|c_\theta\|_\infty}{\varepsilon}\left(\|c_\theta\|_\infty^{-1} c_\theta(x,y) - \|c_\theta\|_\infty^{-1} f_\star(x) - \|c_\theta\|_\infty^{-1} g_\star(y)\right)}.$$

This rescaling allows us to get back to the setup of Rigollet & Stromme (2025), who derive from their assumptions that

$$c_\theta(x,y) \leq 1, \quad f_\star(x) \leq 1 \quad \text{and} \quad g_\star(x) \leq 1,$$

and use these bounds in the rest of their proof. Hence, by rescaling the entropic penalization strength as $\varepsilon^{-1}\|c_\theta\|_\infty$, we obtain

$$e^{-\frac{5\|c_\theta\|_\infty}{\varepsilon}} \leq p_\star(x,y) \leq e^{\frac{5\|c_\theta\|_\infty}{\varepsilon}}$$

from the proof of Proposition 10 of Rigollet & Stromme (2025). We finish the proof similarly as Theorem 6 from Rigollet & Stromme (2025). $\square$

We are now ready to proceed with the proof of Theorem 4.4.

*Proof.* For simplicity, we consider the case where the fairness penalty is independent of the cost function. Using the identity $|a^2 - b^2| \leq |(a+b)||(a-b)|$, we first have that for all $\theta \in \Theta$ that

$$\left| \mathcal{L}_{\mathbf{F}}\left(\mathbf{\Pi}_\epsilon(c_\theta)\right) - \mathcal{L}_{\mathbf{F}}\left(\pi_\epsilon^\star(c_\theta)\right) \right| \lesssim \sum_{s=1}^{K_s} \sum_{w=1}^{K_w} \left| \mathrm{Tr}\left(\mathbf{\Pi}_\epsilon(c_\theta)^\top \mathbf{B}_{sw}\right) - \int_{\mathcal{X} \times \{s\} \times \mathcal{Y} \times \{w\}} d\pi_\epsilon^\star(c_\theta)(x,u,y,t) \right|$$

since

$$\sum_{s=1}^{K_s} \sum_{w=1}^{K_w} \left| \mathrm{Tr}\left(\mathbf{\Pi}_\epsilon^\top(c_\theta) \mathbf{B}_{sw}\right) + \int_{\mathcal{X} \times \{s\} \times \mathcal{Y} \times \{w\}} d\pi_\epsilon^\star(c_\theta)(x,u,y,t) + 2\mathbf{F}_{sw} \right|$$

can be upper-bounded by a constant. Now, for any $(s,w) \in [1, K_s] \times [1, K_w]$ we denote

$$\varphi_{sw} = \xi_{sw} + \mathbf{F}_{sw}$$

where $\xi_{sw}$ is defined by Equation (10). We observe that $\varphi_{sw} \in L^\infty(\mu \otimes \eta)$. Thus, by using Lemma C.4 we have with probability at least $1 - 18e^{-t^2}$

$$\left| \text{Tr}\big(\mathbf{\Pi}_\epsilon^\top(c_\theta)\mathbf{B}_{sw}\big) - \int_{\mathcal{X} \times \{s\} \times \mathcal{Y} \times \{w\}} d\pi_\epsilon^\star(c_\theta)(x, u, y, t) \right| = \left| \int \varphi_{sw} d(\mathbf{\Pi}_\epsilon(c_\theta) - \pi_\epsilon^\star(c_\theta)) \right|$$
$$\lesssim \frac{\exp(5R_\Theta)\epsilon^{-1} \cdot t}{\sqrt{n}} \tag{21}$$

The random variable $X_{sw} := \left| \int \varphi_{sw} d(\mathbf{\Pi}_\epsilon(c_\theta) - \pi_\epsilon^\star(c_\theta)) \right|$ being non-negative, we have

$$\mathbb{E}\big[X_{sw}\big] = \int_0^\infty \mathbb{P}[X_{sw} > u] du.$$

Letting $C_{sw}$ the constant hidden by $\lesssim$ in (21), with the change of variable $u = \frac{C_{sw} \exp(5R_\Theta)\epsilon^{-1} \cdot t}{\sqrt{n}}$, we get

$$\mathbb{E}\big[X_{sw}\big] = \frac{C_{sw} \exp(5R_\Theta)\epsilon^{-1} \cdot}{\sqrt{n}} \int_0^\infty \mathbb{P}\Big[X_{sw} > \frac{C_{sw} \exp(5R_\Theta)\epsilon^{-1} \cdot t}{\sqrt{n}}\Big] dt$$
$$\leq \frac{C_{sw} \exp(5R_\Theta)\epsilon^{-1} \cdot}{\sqrt{n}} \int_0^\infty \min(1, 18e^{-t^2}) dt$$

Let $t_0 = \sqrt{\log(18)}$. It follows

$$\mathbb{E}\big[X_{sw}\big] \leq \frac{C_{sw} \exp(5R_\Theta)\epsilon^{-1} \cdot}{\sqrt{n}} \Big[t_0 + \int_{t_0}^\infty 18e^{-t^2} dt\Big]$$
$$\leq \frac{C_{sw} \exp(5R_\Theta)\epsilon^{-1} \cdot}{\sqrt{n}} \Big[t_0 + \frac{e^{-t_0^2}}{2t_0}\Big]$$
$$\leq \frac{\tilde{C_{sw}} \exp(5R_\Theta)\epsilon^{-1} \cdot}{\sqrt{n}}$$

where the second inequality comes from an integration by parts. Summing over $s$ and $w$ yields for any $\theta \in \Theta$

$$\mathbb{E}[|\mathcal{L}_{\mathbf{F}}(\mathbf{\Pi}_\epsilon(c_\theta) - \mathcal{L}_{\mathbf{F}}(\pi_\epsilon^*(c_\theta))|] \lesssim \frac{\exp(5R_\Theta)\epsilon^{-1}}{\sqrt{n}}$$

As a consequence

$$\sup_{\theta \in \Theta} \mathbb{E}[|\mathcal{L}_{\mathbf{F}}(\mathbf{\Pi}_\epsilon(c_\theta) - \mathcal{L}_{\mathbf{F}}(\pi_\epsilon^*(c_\theta))|] \lesssim \frac{\exp(5R_\Theta)\epsilon^{-1}}{\sqrt{n}}.$$

$\square$

# D. Computational Details

## D.1. Implementation Details

### D.1.1. FAIR SINKHORN

We implement the `FairSinkhorn` algorithm by adapting the Sinkhorn-Knopp (Sinkhorn & Knopp, 1967) implementation of the `Python OT` (POT) library (Flamary et al., 2021; 2024).

### D.1.2. PENALIZED OPTIMAL TRANSPORT

We leverage the POT library (Flamary et al., 2021; 2024) throughout our experiments. To solve the penalized problem, we leverage the `generic conditional gradient` algorithm implemented in this package. We use the following parameters and refer to the package documentations for greater details.

The algorithm is used in conjunction with `line_search_armijo` provided in `Python OT`.

| Parameter | Value |
|---|---|
| G0 | Initialized via a solution of the entropic OT problem |
| numIterMax | 2000 |
| numInnerItermax | 200 |
| stopThr | $10^{-9}$ |
| stopThr2 | $10^{-9}$ |

*Table 1.* Parameters of the `gcg` algorithm used.

### D.1.3. COST LEARNING

We implement bilevel cost learning in Pytorch (Paszke et al., 2019). The inner optimal transport problem is solved using the `ot.sinkhorn` method from `POT`, with a warm-started initialization of the dual variables, a stoping threshold of $10^{-6}$ and a maximum of 1000 iterations. The used solver is `sinkhorn` if the entropic penalty is greater than 1, otherwise we use `sinkhorn_log`. The Jacobian of the optimal transport plan is estimated by iterative differentiation (Pauwels & Vaiter, 2023). We take a single optimizer step after each solve of the inner problem. We found `Adam` (Kingma, 2014) to provide the best and most stable results over runs. We use its default parameters, except for the learning rate. In all experiments, we set the penalty $\mathscr{D}(\cdot, \cdot)$ to be the squared Frobenius norm between the original and parametrized cost matrix.

The Mahalanobis matrix is initialized to the identity, while the MLP is pretrained with fairness penalty $\lambda = 0$ to initialize the model close to the reference cost.

### D.2. Experimental Details on Synthetic Data

In all our experiments involving MLP cost, the MLP model has 2 hidden layers, an output dimension of 2 and uses `ReLU` activation. Moreover, unless otherwise specified, the entropic regularization parameter $\varepsilon$ is set to 1.

To generate results in the top part of Figure 3 b., we generate 250 samples from the group $X$ and 25 from the group $Y$. Half of the sample has sensitive attribute 0, the other half has sensitive attribute 1. For **cost learning** with Mahalanobis cost, we use a penalty grid of `logspace`$(0, 4, 80)$ and the learning rate to 0.1. For **cost learning** with MLP cost, we use a penalty grid of `logspace`$(0, 4, 80)$ and the learning rate to 0.05. For the **penalized OT** results, we use a penalty grid of `logspace`$(0, 3, 80)$. For the **vanilla OT** results, we use an entropic grid of `logspace`$(0, 2, 20)$.

To generate results in the bottom part of Figure 3 b., we generate 250 samples from the group $X$ and 25 from the group $Y$. Half of the sample has sensitive attribute 0, the other half has sensitive attribute 1. For **cost learning** with Mahalanobis cost, we use a penalty grid of `logspace`$(1, 3, 80)$ and the learning rate to 0.05. For **cost learning** with MLP cost, we use a penalty grid of `logspace`$(0, 4, 80)$ and the learning rate to 0.01. For the **penalized OT** results, we use a penalty grid of `logspace`$(0, 3, 80)$. For the **vanilla OT** results, we use an entropic grid of `logspace`$(0, 2, 20)$.

To generate results in Figure 4, we set a fixed entropic penalty of $\varepsilon = 1$. For the penalized approach, the fairness penalty is set to 90. The the cost learning approach with Mahalanobis, the fairness penalty is 1000 and the learning rate is 0.1 while for the MLP, the fairness penalty is 500 and the learning rate is 0.05. The cost are learned using the mixture of Gaussian dataset with 1000 samples from $x$ and 100 samples from $y$. We then evaluate the fairness of each cost model on subsamples of 500 samples from $x$ and 500 samples from $y$. The results are computed for 10 runs.

### D.3. Details on the Dating Data Experiment

**Determine possible matches.** In the dataset, all individuals have a recorded gender (prefer not to say, male, non binary, female or transgender) and sexual orientation (SO). For this last variable, we keep records with values that are one of gay, bisexual, pansexual, lesbian or straight, and remove other records. We then create a compatibility matrix displayed in Figure 6. We acknowledge that this is a simulated setting, and that preferences recorded here might only partially correspond to real-world romantic preferences of individuals.

**Feature selection.** We first remove the following features from the dataset: `app_usage_time_min`, `swipe_right_ratio`, `swipe_right_label`, `mutual_matches`, `profil_pics_count`, `message_sent_count`, `last_active_hour`, `match_outcome`.

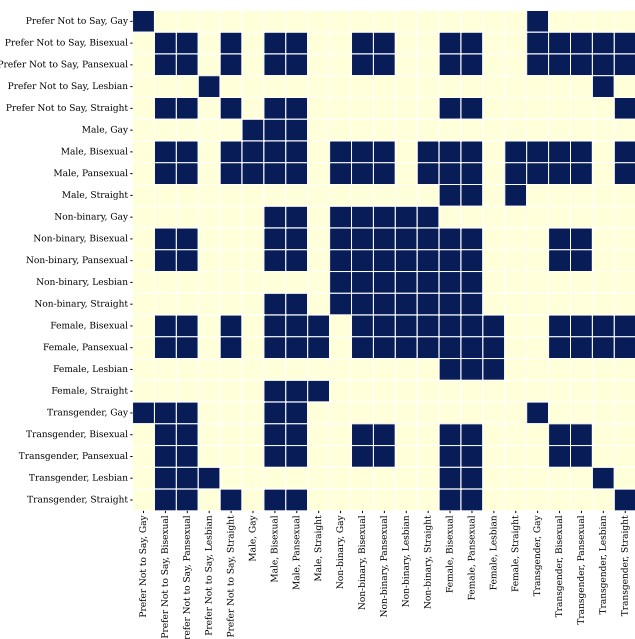

*Figure 6.* Possibles matches in our dating app experiment. Potential matches are displayed in blue, and impossible matches in yellow.

**Realistic Subsampling.** We subsampled the dataset to approximate the contemporary U.S. joint distribution of income bracket and educational attainment. The reference education distribution is based on U.S. Census CPS educational-attainment tables for adults aged 25 and over. The benchmark income distribution is based on a percentile-style mapping of the study's custom income brackets onto U.S. household-income quintiles and the top 5 percent. All details can be found in our Github repository.

**Feature processing.** To map the categorical features encoding income and education level to numerical features, we order them from low to high and then multiply their rank by respectively 20 and 100.

**Distance computation.** First, to account for impossible matches we set the distance between two incompatible individuals to be $10^6$. Our distance function is then computed as a sum of feature-wise distances.

- For interests (f.i. "hiking" and "board games"), we count the number of common interests and then set the distance to be a constant divided by the number of common interests. In our application this constant is equal to 5.

- For location (in our data, metro, rural, small town or urban), app use and swipe time, the distance is equal to 0 if the location match and 5 otherwise.

- For all other numerical variables, the distance is computed as the $\ell_1$ distance between the features.

**Entropic regularization.** The entropic regularization parameter is set to 0.1 for both fair optimal transport approaches.

**Penalized OT.** We use a grid of fairness penalty parameters of `logspace(-1, 2, 17)`. The other parameters of the solver are the same as for the other experiments.

**Cost learning.** The parametrization we use consists of reweighting the feature-wise distances and learning those weights. These weights are initialized to one so that the cost function matches the original cost at initialization. We use a grid of fairness penalty parameters of `logspace(0, 3, 17)`. The learning rate is set to 0.003 and we use the SGD optimizer to learn the cost function.

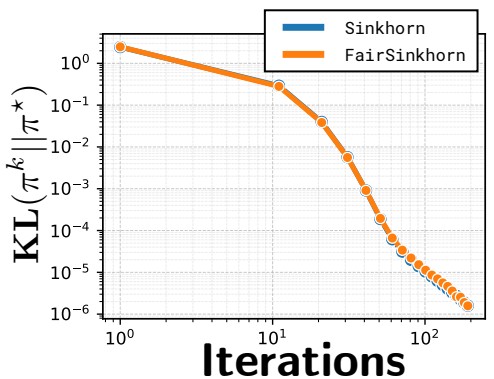
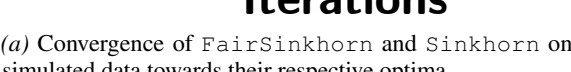
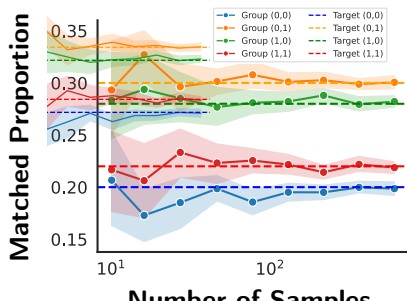

*(a)* Convergence of `FairSinkhorn` and `Sinkhorn` on simulated data towards their respective optima.

*(b)* Results of the deterministic sampling experiment. Dotted lines display the different entries of the fairness target, and solid lines the empirical proportions of matches.

*Figure 7.* Experimental results on simulated data.

### D.4. Additional Experiment on Deterministic Transport Plans

We conduct a supplementary experiment to assess wether sampling from a fair optimal transport plan by row (or column) normalization leads to a transport plan that matches fairness requirements. We use the same data and fairness target as in the Gaussian data experiment, and set the entropic penalty to be $\varepsilon = 1$. We proceed row-wise and sample, for each individual in $\mu_n$, one single match of $\eta_m$. More precisely, given an matrix `ot_plan`, we sample using

$$\texttt{torch.multinomial(ot\_plan, num\_samples=1)}.$$

We average this process over 30 runs with an increasing number of samples and report results in Figure 7b.

### D.5. Experiment on Convergence Speed of `Sinkhorn` and `FairSinkhorn`

We generate data from 5 dimensional Gaussians. Similarly to the previous experiment on Gaussian data, every subgroup has a different center. We use the same fairness target than in other experiments. To obtain the optimal transport plan, we run the Sinkhorn-Knopp algorithm with a maximal number of iterations set to $10^4$ and select the last iteration to be the optimum after checking for convergence. The results are displayed in Figure 7a. One observes that both algorithm achieve similar convergence speed.

