# OpenReview forum: "Optimal Transport under Group Fairness Constraints"
_ICML.cc/2026/Conference — ICML 2026 spotlight_

### Official Review · Reviewer_QWkN · 2026-03-09

**Soundness:** 3
**Presentation:** 3
**Significance:** 3
**Originality:** 3
**Overall Recommendation:** 4
**Confidence:** 2

**Summary:**

In this paper, the authors introduce a new constraint, group fairness, to the Optimal Transport Problem. Specifically, the two populations of the OT Problems are divided into groups, and for any pair of groups, group fairness regulates the percentage of items from one group that are mapped to the other. This paper formalizes this constraint and proves that an OT under it is always achievable under some general assumptions. Further, two approximation algorithms are proposed to control the trade-off between enforcing the constraints and reducing transport costs. The two approaches use different techniques and have respective desirable properties. The results are supported by experiments.

**Compliance With Llm Reviewing Policy:**

Affirmed.

**Final Justification:**

All my concerns are addressed in the rebuttal period, and I would like to keep my current score.

**Key Questions For Authors:**

1. In the paper, currently given a pair of groups $(s,w)$, the fairness constraints specify a number stating how much of group $s$ should be mapped to group $w$. Is it possible to expand this number to a range of ratios, and could the (exact and approximation) algorithms have a better efficiency guarantee uner this relaxed fairness constraints?

**Limitations:**

yes

**Strengths And Weaknesses:**

## Soundness
The submission is technically sound, and the claims are supported by proofs, although I did not thoroughly check the proofs in the Appendix. The theoretical results are supported by the experiments. The evaluation and comparison of different approaches in the article are well illustrated.

## Presentation
The text is well written and structured. The introduction and motivation are natural, the novel notions are easy to digest, and the technical details are not difficult to follow. There are some minor typos in the article, but they generally do not affect the understanding of the content.

- Line 198, Equation (3): The second sum subscript, $w_i$ should be $w_j$
- Line 234, "Interesting" -> "Interestingly"

## Significance
This work shows that group fairness constraints can be applied to OT problems and can be solved efficiently. Further, the fairness constraints are highly customizable, allowing them to be used in many different scenarios, and approximation algorithms are also introduced so that decision makers can customize the trade-off between fairness and efficiency on a case-by-case basis. I believe the work enhances the OT model's capabilities, enabling it to solve a wider range of problems.

## Originality
The introduced concept of group fairness is intuitive and useful, and its combination with OT problems is well articulated and supported. Similar concepts might be able to be applied to other models as well.

---

> ### Author Rebuttal · Authors · 2026-03-31
>
> We thank the reviewer for the positive review and useful feedback.
>
> **Q1: Extending the fairness target to a range**
>
> Thank you for this very interesting question. Let us first consider the exact FairSinkhorn algorithm. In principle, it is possible to replace linear equality constraints with linear range-based constraints and incorporate a projection step onto the new polytope of constraints. A more subtle issue concerns the definition of the fairness target in this setting. In our framework, the fairness target $\mathbf{F}$ is a coupling between the group-wise distributions, which ensures that the available mass for each group is fully distributed across other groups. Hence, for range-based fairness targets, one would need to enforce both that the group-wise ratios lie within the specified intervals *and* that the resulting group-wise matching remains a valid coupling (or alternatively allow some controlled unbalancedness).
>
> For the relaxed algorithms, the solution is more straightforward. Instead of penalizing the squared difference between the observed and target fairness, one could use a margin-based loss that only penalizes violations outside the interval. For example, if the fairness target between groups $(s,w)$ is the interval $[a,b]$, the loss for the transported mass $\pi_{sw}$ could be defined as
> $$\ell(\pi_{sw}) = \max(a - \pi_{sw}, 0) + \max(\pi_{sw} - b, 0),$$
> which is zero when $\pi_{sw}\in[a,b]$ and grows linearly with the violation outside the interval. This formulation is analogous to a hinge loss. Overall, this is a very interesting extension, which we will discuss in the final version of the paper.

---

> > ### Author Rebuttal · Reviewer_QWkN · 2026-04-01
> >
> > Thank the authors for the detailed explanation, and my question is fully addressed.

---

### Official Review · Reviewer_pKUw · 2026-03-12

**Soundness:** 3
**Presentation:** 3
**Significance:** 2
**Originality:** 3
**Overall Recommendation:** 4
**Confidence:** 3

**Summary:**

The authors propose a framework to incorporate group-fairness constraints in Optimal Transport (OT). They present the FairSinkhorn algorithm to enforce exact fairness. Then, they propose two relaxations of the problem, (i) the Fairness-Penalized OT formulation, and (ii) the Fair OT via Cost Learning. (i) is a convex optimization formulation that introduces fairness through a penalty (regularizer) to the OT objective, and has control over the cost-fairness trade-off. (ii) is a bilevel optimization approach that learns a cost function such that the resulting OT plan is fair. Moreover, they provide theoretical guarantees. For (i) they provide a sample complexity bound, and for (ii) a bound of expected fairness deviation. Finally they numerically evaluate their methods on synthetic datasets.

**Compliance With Llm Reviewing Policy:**

Affirmed.

**Key Questions For Authors:**

Questions:

Q1: What are the advantages of Fair OT via Cost Learning over Fairness Penalized OT? Why someone should use it in practice?

Q2: Theorem 4.2 assumes that the cost function is infinitely differentiable. Can this assumption be relaxed? I don't find it common to assume infinite differentiability.

Q3: A time complexity analysis of problems (5) and (7) is absent. Could the authors comment on the practical scalability of the proposed algorithms for large datasets, particularly in comparison to standard entropic optimal transport?

**Limitations:**

Impact Statement: Given that the paper addresses algorithmic fairness and it has direct societal implications, the impact statement would benefit from a more detailed discussion of potential negative societal consequences and limitations.

**Strengths And Weaknesses:**

**Strengths**:

S1: The overall presentation of the paper is well structured, and clearly written. However, some sections are mathematically dense and may be challenging for readers without a strong background in OT.

S2: The problem formulations are well defined, and the proposed relaxations of the exact formulation are meaningful.

S3: The paper consists of a strong and thorough theoretical analysis, supporting the proposed methods.

**Weaknesses:**

W1: The motivation for fairness in optimal transport could be strengthened. While the student allocation example is illustrative, additional real-world scenarios would help clarify the practical relevance of the framework and how it might be deployed in applied settings. Additionally, while the formulation considers fairness in probabilistic transport plans, the target application uses deterministic couplings. It is not clear why it is not more beneficial to directly formulate and solve the fair deterministic transport problem.

W2: Regarding Fairness-Penalized OT:

W2.1: The bounds are well defined mathematically, their quantitative interpretation is not clear. Additional discussion on how these bounds translate into practical performance would improve comprehension. For example, Theorem 4.2 asserts that the sample complexity is the same as entropic OT in terms of the sample order, but in terms of the constant, what is the change?

W2.2: A clearer intuition is required, and a discussion on how these bounds relate to practical fairness scenarios.

W3: Fair OT via Cost Learning:

W3.1: This formulation leads to a non-convex problem that can be difficult to properly optimize.

W3.2: A more detailed ablation study on optimization stability would strengthen the empirical evaluation.

W4: Experimental Evaluation:

W4.1: The experiments are conducted only on synthetic datasets.

W4.2: The absence of real-world evaluations limits the assessment of the method’s practical impact and applicability.

---

> ### Author Rebuttal · Authors · 2026-03-31
>
> We thank the reviewer for the positive feedback and interesting questions. We respond to each point below.
>
> **W1 (part a): Motivation for fairness in optimal transport**
>
> In our response to Reviewer iwxz, we give concrete examples of regulatory rules and legislative goals that can be mapped to the fairness target in our framework. We hope this clarifies the motivation of our work and addresses your concern.
>
> **W1 (part b): Probabilistic vs deterministic couplings**
>
> While our framework is based on probabilistic transport plans, they can be converted to deterministic matchings by sampling from the distributions induced by the coupling (see Remark 3.1). Moreover, in some applications, probabilistic matchings are directly useful. For example, in dating platforms, multiple potential matches can be suggested to a single individual by sampling from the probabilistic coupling, allowing flexibility and diversity in the recommendations.
>
> We have run an experiment to check weather sampling from the probabilistic coupling allows for deterministic matches to be equally fair. The results are available in the **second figure** at this link: https://i.postimg.cc/KcpjGwDG/rebuttal-ot-group-fairness.png. As the number of samples increases, the fairness of the deterministic matching sampled from the probabilistic one converges towards the expected value.
>
> **W2.1: Theorem 4.2 asserts that the sample complexity is the same as entropic OT in terms of the sample order, but in terms of the constant, what is the change?**
>
> First, the main goal of our sample complexity derivation is to provide the well-behaved $\frac{1}{\sqrt{n}}$ scaling. Then, there are two noteworthy dependencies in the constants of our sample complexity upper bound, namely the entropic regularization $\varepsilon$ and the fairness penalty $\lambda$. Our proof proceeds by reducing the sample complexity bound to that of entropic optimal transport. Specifically, we linearize the penalty loss, which results in a shifted cost function $\hat c$ (cf. Equation (14)). Consequently, since the sample complexity grows exponentially with $\lvert c \lvert_{\infty}\varepsilon^{-1}$ (cf. Theorem 3 in Genevay *et al.*, 2019), the scaling on $\varepsilon$ matches that of entropic optimal transport ($e^{\lvert c \lvert_{\infty}\varepsilon^{-1}}$), and yields the following scaling on $\lambda$: $e^{\frac{\lambda K_s K_w}{\varepsilon}}$. That is, for reasonable values of penalization, i.e., $\lambda K_s K_w \lesssim \|c\|_\infty$, the scaling is of the same order as entropic OT.
>
> **W3.1: The cost learning formulation leads to a non-convex problem that can be difficult to properly optimize**
>
> We agree that, due to non-convexity, there is noguarantee of reaching a global minimum with standard optimization algorithms, as noted in Section 4.3. However, we emphasize that the inner problem is strongly convex, and the algorithm we use (gradient descent with iterative differentiation) is guaranteed to converge to a stationary point of the bilevel objective (see Theorem 2 in [1]). Moreover, for a sufficiently small step size, each iteration achieves descent in the objective (see Eq. 40 in [1]).
>
> **W4: Absence of real-world evaluations**
>
> We conducted additional experiments on a dating dataset. Please refer to our response to Reviewer iwxz for details. The results are available at the following link in the **first figure**: https://i.postimg.cc/KcpjGwDG/rebuttal-ot-group-fairness.png. We hope this addresses your concern.
>
> **Q1: Advantages of cost learning over fairness penalized OT**
>
> We discuss this question in detail in Section 4.3 and Figure 4. The key advantage of the cost learning approach is that it yields a reusable cost-function: or any new set of individuals, one can simply solve the standard optimal transport problem with the learned cost, without retraining. As shown in Figure 4, this results in a substantial computational benefit compared to penalized OT, achieving comparable levels of fairness with roughly two orders of magnitude less computation time.
>
> **Q2: Assumption of infinite differentiability**
>
> We note that many cost functions used in practice satisfy this requirement. For example, cost functions of the form $||\cdot||_p^p$ with even $p \geq 2$, their weighted counterparts, and cost functions parameterized by smooth neural networks are all infinitely differentiable. To the best of our knowledge, existing results on the sample complexity of optimal transport also rely on this assumption [2,3].
>
> --------------
> ### References
>
> [1] [Bilevel Optimization: Convergence Analysis and Enhanced Design, Ji et al., 2021](https://arxiv.org/abs/2010.07962)
>
> [2] [Sample Complexity of Sinkhorn divergences, Genevay et al., 2018](https://arxiv.org/abs/1810.02733)
>
> [3] [Statistical bounds for entropic optimal transport: sample complexity and the central limit theorem, Mena et al., 2021](https://arxiv.org/abs/1905.11882)

---

> > ### Author Rebuttal · Reviewer_pKUw · 2026-04-02
> >
> > I thank the authors for their response, which has largely addressed my questions. I hope that the authors can include some of these discussions in the revised version.

---

> > > ### Author Response · Authors · 2026-04-02
> > >
> > > Dear Reviewer,
> > >
> > > Thank you for your response. We will make sure to incorporate all of these elements into the revised version of the manuscript, and we greatly appreciate your comments, which have helped us improve the manuscript.
> > >
> > > As our revisions appear to have addressed your concerns, we hope that you might consider increasing your score to 5.

---

### Official Review · Reviewer_1wKB · 2026-03-13

**Soundness:** 3
**Presentation:** 4
**Significance:** 4
**Originality:** 3
**Overall Recommendation:** 5
**Confidence:** 4

**Summary:**

The proposes a variant of OT for group fairness for applications where the target fairness matrix, containing the desired levels of coupling probability, is available. The paper mainly proposes two approaches FairSinkhorn - a modification of the Sinkhorn algorithm to enforce exact fairness constraints and a bilevel optimization approach to learn a ground cost that induces a fair solution. The finite sample complexity rate of O(m^{-1/2}) is derived both for the entropic OT case & the bilevel optimization of learning the cost function first. Empirical experiments illustrate the theory.

**Compliance With Llm Reviewing Policy:**

Affirmed.

**Final Justification:**

The paper proposes a novel notion of group fairness in optimal transport (OT) & demonstrates a Sinkhorn-based solver & also obtains favorable sample complexity. The method is conceptually simple yet practically relevant and it's presented well.

The authors have agreed to include and discuss the suggested references on structured OT and fairness in OT, which should improve the positioning of the paper.

The authors have added additional experiments on a real-world dataset and provided further empirical analysis, including results on deterministic couplings and runtime overhead. While these additions strengthen the empirical section, I still believe that evaluating on a wider range of large-scale or downstream tasks would further solidify the claims, particularly regarding the effect of relaxing metric properties.

Overall, the rebuttal adequately addresses most of my concerns and makes technically sound contribution to fairness in OT.

**Key Questions For Authors:**

1. For converting the probabilistic coupling to deterministic coupling, barycentric projection is the most common approach. There could have been a discussion on how well barycentric projection performs on retaining the fair couplings learned by the probabilistic map.

2. There should be an empirical comparison of the extra runtime that the projection step brings in the Sinkhorn algorithm and of sensitivity of the hyperparameters.

**Limitations:**

There is no discussion on limitations. There is only a mention of future work to evaluate the approach on downstream applications.

**Strengths And Weaknesses:**

**Strengths**

While most prior works either use OT to ensure fairness or work with the individual fairness notion for OT. The proposed work proposes a novel notion of group fairness in OT.

The algorithm to solve can be simply modified from the computationally efficient Sinkhorn algorithm with an extra projection step.
While the proof techniques might not be new, the sample complexity bounds are favourable.

The paper is overall well written, except for some mixup of citet & citep and the notation of B introduced in proposition 3.3 without possibly being defined.

**Weaknesses**

The framework relies on the availability of the groun truth fairness matrix F. With the availability of F, the proposed approach is quiet similar to what happens in the explainability literature where the deviation from ground truth saliecy maps is penalized. However, the Sinkhorn-based scalability is still a notable advantage of the proposed approach.

Some related works are missing: (1) Structured Optimal Transport (Alvarez-Melis et. al, 2017) which could be compared against for learning submodular (structured) cost for OT (2) Submodular framework for structured-sparse optimal transport (Manupriya et. al, 2024) related to learning structured sparse transport plan (3) Equitable and Optimal Transport with Multiple Agents (Scetbon et. al, 2021) related to a different notion of fairness in OT

The experimental results could've been on more large scale datasets or wider applications to check if the loss of metric properties affects the downstream applications.

---

> ### Author Rebuttal · Authors · 2026-03-31
>
> We thank the reviewer for the positive and constructive feedback.
>
> **Connection to explainability literature**
>
> Thank you for pointing out this interesting connection. We will discuss it in the revised version of the paper, with explicit references to [1, 2].
>
> **Additional related work in OT**
>
> Thank you for providing these additional references on optimal transport. We will include [3, 4] in the related work section to highlight relevant connections with structured OT, and add [5] to the paragraph on fairness in OT.
>
> **Additional experimental results**
>
> Following your request, we have conducted additional experiments on a real-world dating dataset; details are provided in our response to Reviewer iwxz.
>
> The results are available in the first figure at this link: https://i.postimg.cc/KcpjGwDG/rebuttal-ot-group-fairness.png
>
> **Q1: Converting probabilistic couplings to deterministic couplings**
>
> Barycentric projection is most natural when the transport plan is interpreted as a map between distributions over a continuous domain. In our setting, however, the goal is to produce a matching between observed individuals, so the barycenter does not, in general, correspond to an observed individual. While one could resort to heuristics such as projecting on the nearest neighbor to the barycenter, we instead propose a sampling-based procedure described in Remark 3.1. This approach directly produces feasible matchings that, for large enough groups, concentrate towards a fair deterministic matching.
>
> To check this, we have run a supplementary experiment in which we sample from the probabilistic coupling. The results are available at this link in the **second figure**: https://i.postimg.cc/KcpjGwDG/rebuttal-ot-group-fairness.png. As the number of individuals to match grows, the fairness of the deterministic coupling sampled from the probabilistic one converges towards the fairness target.
>
> **Q2: Empirical comparison of extra runtime of projection step in Sinkhorn**
>
> This is indeed a relevant question. We have run FairSinkhorn and regular Sinkhorn over 10 iterations. The runtimes are reported in the **third figure** at the following link: https://i.postimg.cc/KcpjGwDG/rebuttal-ot-group-fairness.png
>
>
> ----------
> References
>
> [1] [End-to-End Saliency Mapping via Probability Distribution Prediction, Jetley et al., 2018](https://arxiv.org/abs/1804.01793)
>
> [2] [Towards Fairness for the Right Reasons: Using Saliency Map to Evaluate Bias Removal in Neural Networks, Sztukiewicz et al., 2025](https://arxiv.org/abs/2503.00234v1)
>
> [3] [Structured Optimal Transport, Alvarez-Melis et. al, 2017](https://arxiv.org/abs/1712.06199)
>
> [4] [Submodular Framework for Structured-Sparse Optimal Transport, Manupriya et. al, 2024](https://arxiv.org/abs/2406.04914)
>
> [5] [Equitable and Optimal Transport with Multiple Agents, Scetbon et. al, 2021](https://arxiv.org/abs/2006.07260)

---

> > ### Author Rebuttal · Reviewer_1wKB · 2026-04-02
> >
> > Thank you for the response and the additional experiment. I would like to stay with the positive score.

---

### Official Review · Reviewer_iwxz · 2026-03-13

**Soundness:** 3
**Presentation:** 3
**Significance:** 3
**Originality:** 3
**Overall Recommendation:** 5
**Confidence:** 3

**Summary:**

This submission introduces a novel notion of group fairness for optimal transport (OT) and proposes a modified *Sinkhorn* algorithm. To degrade matching quality in practice, this paper develops two relaxation strategies: *Fairness-Penalized OT* and *Fair OT via cost learning*. Both theoretical guarantees and empirical evaluations are provided to demonstrate the correctness of the proposed approach.

**Compliance With Llm Reviewing Policy:**

Affirmed.

**Final Justification:**

All of my concerns have been addressed, and I am increasing my overall rating to 5.

**Key Questions For Authors:**

Should the authors adequately address **W1**, I would be inclined to raise my score.

**Limitations:**

yes

**Strengths And Weaknesses:**

**Strength**
* S1. Important and meaningful problem. This paper addresses the issue of group fairness, which is of significant practical importance in modern society, particularly in domains such as resource allocation, job recommendation, and university admissions. The authors clearly explain the potential for unfair outcomes arising from traditional matching mechanisms.
* S2. Novel definition. This paper introduces a novel definition of group fairness, which is well-suited for studying fairness problems within the optimal transport framework.
* S3. Detailed theoretical analysis. This paper provides new finite-sample complexity bounds for entropy-regularized optimal transport with fairness penalties, demonstrating that incorporating fairness constraints does not compromise statistical efficiency. It also establishes fairness bias bounds for cost-learning methods, ensuring that the learned cost functions generalize effectively to unseen samples.

**Weaknesses**
* W1. Dataset concern. Although the introduction mentions examples such as university admissions and dating applications, the experiments are conducted solely on synthetic data.
Evaluating the method on real-world datasets would further enhance the paper’s practical significance and more clearly illustrate its potential for addressing fairness challenges in real-world applications.
* W2. Lack of comparative baselines. More references to state-of-the-art optimal transport (OT) methods should be included to strengthen the paper’s credibility and persuasive power.
* W3. It is unclear how the fairness target (Line 347) is determined. The authors should further clarify whether references support this setup or have practical relevance.

---

> ### Author Rebuttal · Authors · 2026-03-31
>
> We thank you for the positive and constructive feedback. We address your questions below.
>
> **W1. Evaluation on real-world data**
>
> Thank you for your suggestion. To address your request, we conducted experiments on the Dating App Behavior Dataset from Kaggle [1]. This dataset provides individual-level features but does not include observed matches between individuals, requiring us to design an ad hoc cost function (see below). If you are aware of any dataset that combines both rich feature information and observed matches, we would be very interested in incorporating it into the camera-ready version of the paper.
>
> To enhance realism, we subsampled the original data such that key variables (such as education level and income) match the distribution reported in US Census data. We also accounted for the diversity of sexual orientations, dating preferences, and gender identities by constructing a match compatibility matrix that specifies feasible matches (e.g., two gay men can match, where two heterosexual women cannot).
>
> As a baseline, we defined a simple ad hoc cost function that measures similarity between individuals based on reported characteristics Assuming individuals are described by $d$ features, the baseline cost between two individuals $x_i$ and $y_j$ takes the simple form $\sum\limits_{k=1}^d d_k(x_i^{(d)},y_j^{(d)}).$ For categorical variables (except income level and education level), we define the distance $d_k$ to be $0$ if their value matches, and $c >0$ otherwise, and for numerical values we compute their absolute difference. We map education and income levels to a discrete set of numbers to account for their ordinal structure (e.g., low income is closer to low-middle than to high), and then compute their absolute differences. This custom distance is similar in spirit to Gower's distance. We provide a visualization of the possible matching matrix, the distance matrix on a subsample of individuals, and the distribution of the values of the distance at the link attached to this comment. For our cost learning approach, we simply learn weights $\alpha_k$ over the feature-level distances $d_k$. This ensures that the resulting cost function remains highly interpretable.
>
> The results are given in the **first figure** provided at this link: https://i.postimg.cc/KcpjGwDG/rebuttal-ot-group-fairness.png
>
> We observe a similar fairness v.s. cost difference trade-off as the one observed on synthetic experiments.
>
> **W2. Lack of comparative baselines**
>
> We would be happy to expand the discussion of related work if the reviewer can suggest for relevant methods or references. However, we would like to emphasize that, to the best of our knowledge, our work is the first to address group fairness in optimal transport. As such, this problem setting is novel and does not yet admit established baselines for direct comparison.
>
> **W3. How the fairness target is determined**
>
> Thank you for this question. The fairness target is intended to be specified by a policymaker or regulator and is inherently context-dependent. It should be interpreted as a representation of proportionality constraints that are commonly observed in practice, including employment law, affirmative action policies with explicit numerical targets, and quota-based systems in education and public governance.
>
> We provide three illustrative examples. First, in the US, hiring processes are evaluated under the four-fifths rule, which requires that the selection rate of any demographic group should be at least 80% of that of the group with the highest selection rate [2]. Second, in the French higher education system, several institutions are subject to minimum quotas for scholarship holders [3]. Third, at the European level, a recent EU Directive sets a target whereby listed companies should ensure that at least 40% of executive non-executive director positions, or 33% of all director positions, are held by women by 2026 [4]. Such rules or legislative goals can be naturally mapped to the fairness target in our framework.
>
> We will incorporate these examples in the revised version of the paper.
>
> ----------
> References
>
> [1] https://www.kaggle.com/datasets/keyushnisar/dating-app-behavior-dataset/
>
> [2] https://en.wikipedia.org/wiki/Disparate_impact
>
> [3] https://draaf.hauts-de-france.agriculture.gouv.fr/parcoursup-2025-encourager-la-mobilite-sociale-et-geographique-sur-parcoursup-a3979.html (source in French)
>
> [4] https://eur-lex.europa.eu/eli/dir/2022/2381/oj/eng

---

> > ### Author Rebuttal · Reviewer_iwxz · 2026-04-01
> >
> > I appreciate your rebuttal, which has addressed my concerns. I will revise my rating accordingly.

---

### Decision · Program_Chairs · 2026-04-30

**Decision:**

Accept (spotlight)

**Comment:**

The submission considers the problem of algorithmic matching under fairness constraints. Specifically, the authors focus on optimal transport (OT) problems and introduce notions of group fairness tailored for OT plans. They propose FairSinkhorn, an algorithm that aims to compute the fairness-constrained optimal transport efficiently. Noting that the resulting solutions may incur significant distortion, they propose two relaxations of their formulation and test the empirical performance of their methods on numerical experiments.

The reviewers were unanimous in recommending acceptance and recognizing the quality of the work. Reviewer iwxz increased their rating post-rebuttal, noting that all their concerns were addressed. They had raised questions about the dataset and the lack of comparative baselines, which the authors addressed by adding experiments on a real-world dating dataset and by noting that their work is the first to address group fairness in optimal transport.  Reviewer 1wKB also had a favorable opinion, noting that the rebuttal adequately addressed their concerns, particularly regarding related work on structured OT and fairness in OT. Similarly, Reviewer pKUw had a favorable opinion of the paper, acknowledging that the authors had largely addressed their concerns. Finally, reviewer QWkN did not raise serious issues with the paper.

This is a nice contribution to the literature on fair machine learning, and the optimal transport focus, the introduction of the FairSinkhorn algorithm, ant the theoretical analysis are particularly interesting. The paper provides an (increasingly rare) combination of theoretical contributions, an interesting formulation for a real problem, and compelling experiments. It will be a welcome addition to the ICML program.